# Discovering and Achieving Goals via World Models

**Russell Mendonca***
Carnegie Mellon University

**Oleh Rybkin***
University of Pennsylvania

**Kostas Daniilidis**
University of Pennsylvania

**Danijar Hafner**
University of Toronto

**Deepak Pathak**
Carnegie Mellon University

## Abstract

How can artificial agents learn to solve many diverse tasks in complex visual environments without any supervision? We decompose this question into two challenges: discovering new goals and learning to reliably achieve them. Our proposed agent, Latent Explorer Achiever (LEXA), addresses both challenges by learning a world model from image inputs and using it to train an explorer and an achiever policy via imagined rollouts. Unlike prior methods that explore by reaching previously visited states, the explorer plans to discover unseen surprising states through foresight, which are then used as diverse targets for the achiever to practice. After the unsupervised phase, LEXA solves tasks specified as goal images zero-shot without any additional learning. LEXA substantially outperforms previous approaches to unsupervised goal reaching, both on prior benchmarks and on a new challenging benchmark with 40 test tasks spanning across four robotic manipulation and locomotion domains. LEXA further achieves goals that require interacting with multiple objects in sequence.

## 1   Introduction

How can we build an agent that learns to solve hundreds of tasks in complex visual environments, such as rearranging objects with a robot arm or completing chores in a kitchen? While traditional reinforcement learning (RL) has been successful for individual tasks, it requires a substantial amount of human effort for every new task. Specifying task rewards requires domain knowledge, access to object positions, is time-consuming, and prone to human errors. Moreover, traditional RL would require environment interaction to explore and practice in the environment for every new task. Instead, we approach learning hundreds of tasks through the paradigm of unsupervised goal-conditioned RL, where the agent learns many diverse skills in the environment in the complete absence of supervision, to later solve tasks via user-specified goal images immediately without further training [2, 26, 40].

**Challenges**   Exploring the environment and learning to solve many different tasks is substantially more challenging than traditional RL with a dense reward function or learning from

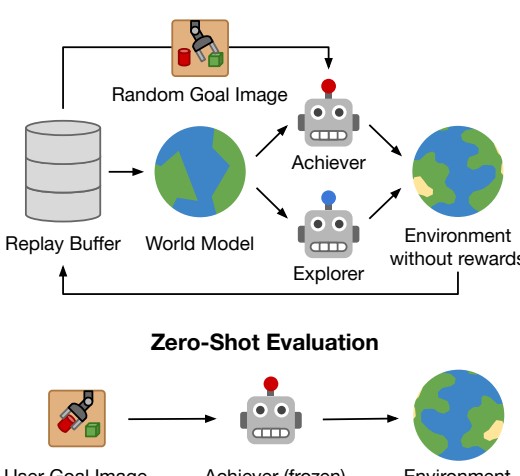

Figure 1: LEXA learns a world model without any supervision, and leverages it to train two policies in imagination. The *explorer* finds new images and the *achiever* learns to reliably reach them. Once trained, the achiever reaches user-specified goals zero-shot without further training at test time.

---

* Equal contribution. Ordering determined at random. Project page: https://orybkin.github.io/lexa/

35th Conference on Neural Information Processing Systems (NeurIPS 2021).

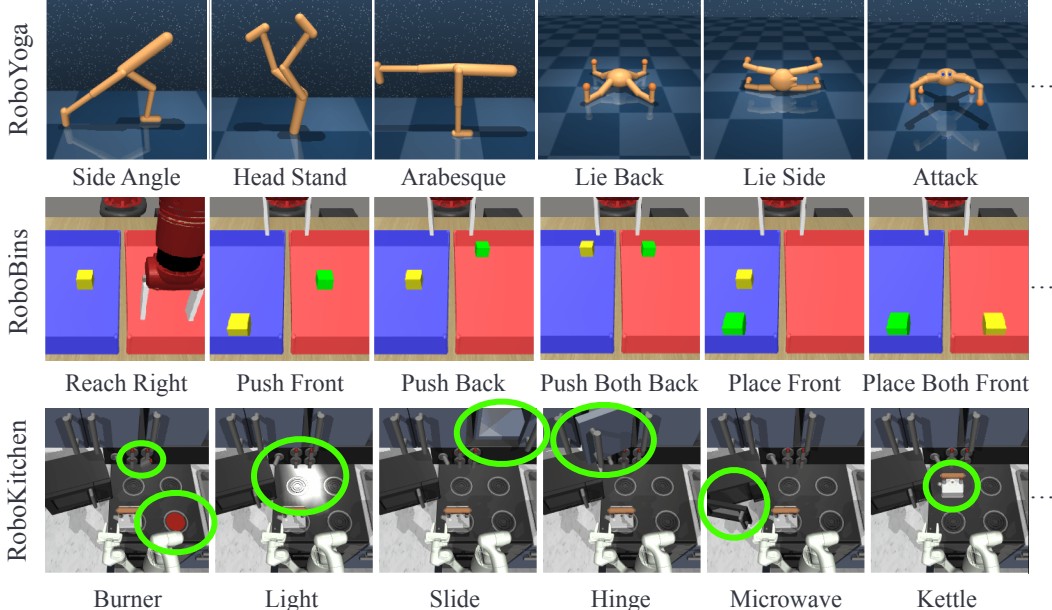

Figure 2: We benchmark LEXA across four visual control environments. A representative sample of the test-time goals is shown here. RoboYoga features complex locomotion and precise control of high-dimensional agents, RoboBins manipulation with multiple objects, and RoboKitchen a variety of diverse tasks that require complex control strategies such as opening a cabinet.

expert demonstrations. Existing methods are limited to simple tasks, such as picking or pushing a puck [13, 32, 37] or controlling simple 2D robots [50]. The key challenge in improving the performance of unsupervised RL is *exploration*. In particular, previous approaches explore by either revisiting previously seen rare goals [14, 18, 55] or sampling goals from a generative model [32, 37]. However, in both these approaches, the policy as well as the generative model are trained on previously visited states from the replay buffer, and hence the sampled goals are either within or near the frontier of agent's experience. Ideally, we would like the agent to discover goals much beyond its frontier for efficient exploration, but how does an agent generate goals that it is yet to encounter? This is an open question not just for AI but for cognitive science too [42].

**Approach**    To rectify this issue, we leverage a learned world model to train a separate *explorer* and *achiever* policy in imagination. Instead of randomly sampling or generating goals, our explorer policy discovers distant goals by first planning a sequence of actions optimized in imagination of the world model to find novel states with high expected information gain [30, 43, 44]. It then executes those imagined actions in the environment to discover interesting states without the need to generate them. Note these actions are likely to lead the agent to states which are several steps outside the frontier because otherwise the model wouldn't have had high uncertainty or information gain. Finally, these discovered states are used as diverse targets for the achiever to practice. We train the achiever from on-policy imagination rollouts within the world model and without relying on experience relabeling, therefore leveraging *foresight over hindsight*. After this unsupervised training phase, the achiever solves tasks specified as goal images zero-shot without any additional learning at deployment. Unlike in the conventional RL paradigm [31, 47], our method is trained once and then used to achieve several tasks at test time without any supervision during training or testing.

**Contributions**    We introduce Latent Explorer Achiever (LEXA), an unsupervised goal reaching agent that trains an explorer and an achiever within a shared world model. At training, LEXA unlocks diverse data for goal reaching in environments where exploration is nontrivial. At test time, the achiever solves challenging locomotion and manipulation tasks provided as user-specified goal images. Our contributions are summarized as follows:

- We propose to learn separate explorer and achiever policies as an approach to overcome the exploration problem of unsupervised goal-conditioned RL.
- We show that forward-looking exploration by planning with a learned world model substantially outperforms previous strategies for goal exploration.
- To evaluate on challenging tasks, we introduce a new goal reaching benchmark with a total of 40 diverse goal images across 4 different robot locomotion and manipulation environments.
- LEXA outperforms prior methods, being the first to show success in the Kitchen robotic manipulation environment, and achieves goal images where multiple objects need to be moved.

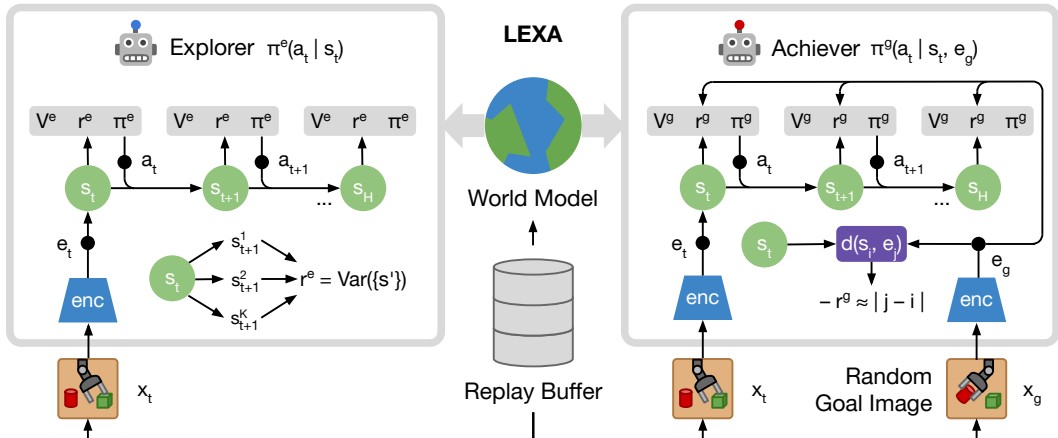

Figure 3: Latent Explorer Achiever (LEXA) learns a general world model that is used to train an explorer and a goal achiever policy. The explorer (left) is trained on imagined latent state rollouts of the world model $s_{t:T}$ to maximize the disagreement objective $r_t^e = \mathrm{Var}(s')$. The goal achiever (right) is conditioned on a goal $g$ and is also trained on imagined rollouts to minimize a distance function $d(s_t, e_g)$. Goals are sampled randomly from replay buffer images. For training a temporal distance, we use the imagined rollouts of the achiever and predict the number of time steps between each two states. By combining forward-looking exploration and data-efficient training of the achiever, LEXA provides a simple and powerful solution for unsupervised reinforcement learning.

## 2   Latent Explorer Achiever (LEXA)

Our aim is to build an agent that can achieve arbitrary user-specified goals after learning in the environment without any supervision. This presents two challenges - collecting trajectories that contain diverse goals and learning to achieve these goals when specified as a goal image. We introduce a simple solution based on a world model and imagination training that addresses both challenges. The world model represents the agent's current knowledge about the environment and is used for training two policies, the explorer and the achiever. To explore novel situations, we construct an estimate of which states the world model is still uncertain about. To achieve goals, we train the goal-conditioned achiever in imagination, using the images found so far as unsupervised goals. At test time, the achiever is deployed to reach user-specified goals. The training procedure is in Algorithm 1.

### 2.1   World Model

To efficiently predict potential outcomes of future actions in environments with high-dimensional image inputs, we leverage a Recurrent State Space Model (RSSM) [23] that learns to predict forward using compact model states that facilitate planning [7, 51]. In contrast to predicting forward in image space, the model states enable efficient parallel planning with a large batch size and can reduce accumulating errors [39]. The world model consists of the following components:

$$
\begin{array}{llll}
\text{Encoder:} & e_t = \mathrm{enc}_\phi(x_t) & \text{Posterior:} & q_\phi(s_t \mid s_{t-1}, a_{t-1}, e_t) \\
\text{Dynamics:} & p_\phi(s_t \mid s_{t-1}, a_{t-1}) & \text{Image decoder:} & p_\phi(x_t \mid s_t)
\end{array}
\tag{1}
$$

The model states $s_t$ contain a deterministic component $h_t$ and a stochastic component $z_t$ with diagonal-covariance Gaussian distribution. $h_t$ is the recurrent state of a Gated Recurrent Unit (GRU) [11]. The encoder and decoder are convolutional neural networks (CNNs) and the remaining components are multi-layer perceptrons (MLPs). The world model is trained end-to-end by optimizing the evidence lower bound (ELBO) via stochastic backpropagation [28, 38] with the Adam optimizer [27].

### 2.2   Explorer

To efficiently explore, we seek out surprising states imagined by the world model [6, 41, 43, 44, 46], as opposed to retrospectively exploring by revisiting previously novel states [4, 5, 8, 34]. As the world model can predict model states that correspond to unseen situations in the environment, the imagined trajectories contain more novel goals, compared to model-free exploration that is limited to the replay buffer. To collect informative novel trajectories in the environment, we train an exploration

---

**Algorithm 1:** Latent Explorer Achiever (LEXA)

1: **initialize:** World model $\mathcal{M}$, Replay buffer $\mathcal{D}$, Explorer $\pi^{\mathrm{e}}(a_t \mid z_t)$, Achiever $\pi^{\mathrm{g}}(a_t \mid z_t, g)$
2: **while** exploring **do**
3:     Train $\mathcal{M}$ on $\mathcal{D}$
4:     Train $\pi^{\mathrm{e}}$ in imagination of $\mathcal{M}$ to maximize exploration rewards $\sum_t r_t^{\mathrm{e}}$.
5:     Train $\pi^{\mathrm{g}}$ in imagination of $\mathcal{M}$ to maximize $\sum_t r_t^{\mathrm{g}}(z_t, g)$ for images $g \sim \mathcal{D}$.
6:     (Optional) Train $d(z_i, z_j)$ to predict distances $j - i$ on the imagination data from last step.
7:     Deploy $\pi^{\mathrm{e}}$ in the environment to explore and grow $\mathcal{D}$.
8:     Deploy $\pi^{\mathrm{g}}$ in the environment to achieve a goal image $g \sim \mathcal{D}$ to grow $\mathcal{D}$.
9: **end while**

10: **while** evaluating **do**
11:     **given:** Evaluation goal $g$
12:     Deploy $\pi^{\mathrm{g}}$ in the world to reach $g$.
13: **end while**

---

policy $\pi^e$ from the model states $s_t$ in imagination of the world model to maximize an exploration reward:

$$\text{Explorer:} \qquad \pi^e(a_t \mid s_t) \qquad \text{Explorer Value:} \qquad v^e(s_t) \tag{2}$$

To explore the most informative model states, we estimate the epistemic uncertainty as a disagreement of an ensemble of transition functions. We train an ensemble of 1-step models to predict the next model state from the current model state. The ensemble model is trained alongside the world model on model states produced by the encoder $q_\phi$. Because the ensemble models are initialized at random, they will differ, especially for inputs that they have not been trained on [29, 36]:

$$\text{Ensemble:} \quad f(s_t, \theta^k) = \hat{z}_{t+1}^k \quad \text{for} \quad k = 1..K \tag{3}$$

Leveraging the ensemble, we estimate the epistemic uncertainty as the ensemble disagreement. The exploration reward is the variance of the ensemble predictions averaged across dimension of the model state, which approximates the expected information gain [3, 43]:

$$r_t^{\mathrm{e}}(s_t) \doteq \frac{1}{N} \sum_n \mathrm{Var}_{\{k\}} \left[ f(s_t, \theta_k) \right]_n \tag{4}$$

The explorer $\pi^e$ maximizes the sum of future exploration rewards $r_t^e$ using the Dreamer algorithm [24], which considers long-term rewards into the future by maximizing $\lambda$-returns under a learned value function. As a result, the explorer is trained to seek out situations are as informative as possible from imagined latent trajectories of the world model, and is periodically deployed in the environment to add novel trajectories to the replay buffer, so the world model and goal achiever policy can improve.

## 2.3 Achiever

To leverage the knowledge obtained by exploration for learning to reach goals, we train a goal achiever policy $\pi^g$ that receives a model state and a goal as input. Our aim is to train a general policy that is capable of reaching many diverse goals. To achieve this in a data-efficient way, it is crucial that environment trajectories that were collected with one goal in mind are reused to also learn how to reach other goals. While prior work addressed this by goal relabeling which makes off-policy policy optimization a necessity [2], we instead leverage past trajectories via the world model trained on them that lets us generate an unlimited amount of new imagined trajectories for training the goal achiever on-policy in imagination. This simplifies policy optimization and can improve stability, while still sharing all collected experience across many goals.

$$\text{Achiever:} \qquad \pi^g(a_t \mid s_t, e_g) \qquad \text{Achiever Value:} \qquad v^g(s_t, e_g) \tag{5}$$

To train the goal achiever, we sample a goal image $x_g$ from the replay buffer and compute its embedding $e_g = \mathrm{enc}_\phi(x_g)$. The achiever aims to maximize an unsupervised goal-reaching reward $r^g(s_t, e_g)$. We discuss different choices for this reward in Section 2.4. We again use the Dreamer algorithm [24] for training, where now the value function also receives the goal embedding as input.

In addition to imagination training, it can also be important to perform practice trials with the goal achiever in the true environment, so that any model inaccuracies along the goal reaching trajectories

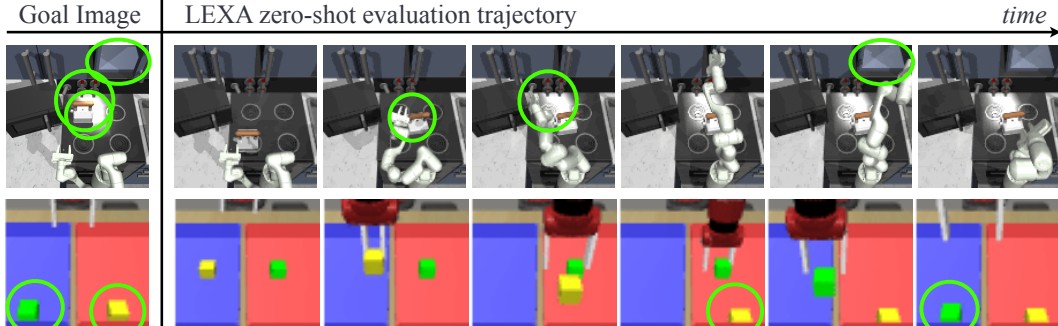

| Goal Image | LEXA zero-shot evaluation trajectory | *time* |

Figure 4: Successful LEXA trajectories. When given a goal image from the test set, LEXA's achiever is used in the environment to reach that image. On RoboKitchen, LEXA manipulates up to three different objects together from a single goal image (kettle, light switch, and cabinet). On RoboBins, LEXA performs temporally extended tasks such as picking and placing two objects in a row.

may be corrected. To perform practice trials, we sample a goal from the replay buffer and execute the goal achiever policy for that goal in the environment. These trials are interleaved with exploration episodes collected by the exploration policy in equal proportion. We note that the goal achiever learning is entirely unsupervised because the practice goals are simply images the agent encountered through exploration or during previous practice trails.

## 2.4 Latent Distances

Training the achiever policy requires us to define a goal achievement reward $r^g(s_t, e_g)$ that measures how close the latent state $s_t$ should be considered to the goal $e_g$. One simple measure is the cosine distance in the latent space obtained by inputting image observations into the world-model. However, such a distance function brings *visually* similar states together even if they could be farther apart in *temporal* manner as measured by actions needed to reach from one to other. This bias makes this suitable only to scenarios where most of pixels in the observations are directly controllable, e.g., trying to arrange robot's body in certain shape, such as RoboYoga poses in Figure 2. However, many environments contain agent as well as the world, such as manipulation involves interacting with objects that are not directly controllable. The cosine distance would try matching the entire goal image, and thus places a large weight on both matching the robot and object positions with the desired goal. Since the robot position is directly controllable it is much easier to match, but this metric overly focuses on it, yielding poor policies that ignore objects. We address this is by using the number of timesteps it takes to move from one image to another as a distance measure [25, 26]. This ignores large changes in robot position, since these can be completed in very few steps, and will instead focus more on the objects. This temporal cost function can be learned purely in imagination rollouts from our world model allowing as much data as needed without taking any steps in the real world.

**Cosine Distance**   To use cosine distance with LEXA, for a latent state $s_t$, and a goal embedding $e^g$, we use the latent inference network $q$ to infer $s^g$, and define the reward as the cosine similarity [54]:

$$r^g_t(s_t, e_g) \doteq \sum_i \overline{s}_{ti}\overline{s}_{gi}, \quad \text{where} \quad \overline{s}_t = s_t/\|s_t\|_2, \quad \overline{s}_g = s_g/\|s_g\|_2, \tag{6}$$

i.e. the cosine of the angle between the two vectors $s_t, s_g$ in the $N-$dimensional latent space.

**Temporal Distance**   To use temporal distances with LEXA, we train a neural network $d$ to predict the number of time steps between two embeddings. We train it by sampling pairs of states $s_t, s_{t+k}$ from an imagined rollout of the achiever and predicting the distance $k$. We implement the temporal distance in terms of predicted image embeddings $\hat{e}_{t+k}$ in order to remove extra recurrent information:

$$\text{Predicted embedding:} \quad \text{emb}(s_t) = \hat{e}_t \approx e_t \quad \text{Temporal distance:} \quad d_\omega(\hat{e}_t, \hat{e}_{t+k}) \approx k/H, \tag{7}$$

where $H$ is the maximum distance equal to the imagination horizon. Training distance function only on imagination data from the same trajectory would cause it to predict poor distance to far away states coming from other trajectories, such as images that are impossible to reach during one episode. In order to incorporate learning signal from such far-away goals, we include them by sampling images

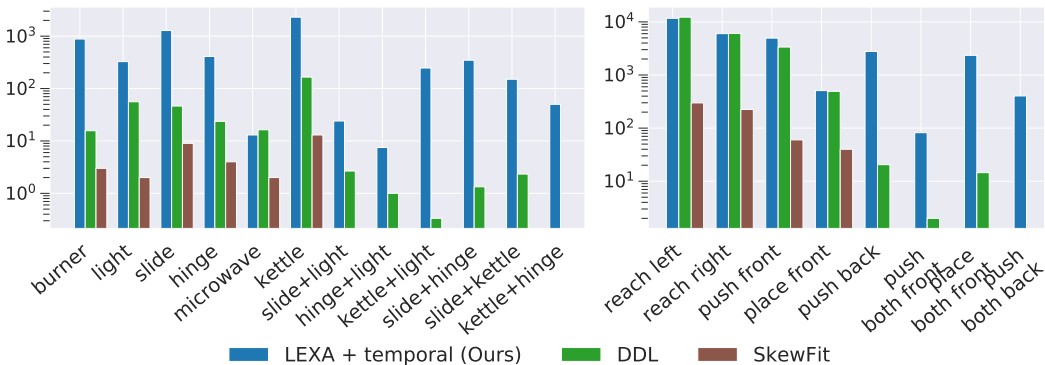

Figure 5: Coincidental goal success achieved during the unsupervised exploration phase. The forward-looking explorer policy of LEXA results in substantially better coverage compared to SkewFit, a popular method for goal based exploration.

from a different trajectory. We annotate these negative samples with the maximum possible distance, so that the agent always prefers images that were seen in the same trajectory.

$$r_t^g(s_t, e_g) = -d_\omega(\hat{e}_t, e_g), \quad \text{where} \quad \hat{e}_t = \text{emb}(s_t), \quad e_g = \text{enc}_\phi(x_g) \tag{8}$$

The learned distance function depends on the training data policy. However, as the policy becomes more competent, the distance estimates will be closer to the optimal number of time steps to reach a particular goal, and the policy converges to the optimal solution [25]. LEXA always uses the latest data to train the distance function using imagination, ensuring that the convergence is fast.

## 3 Experiments

Our evaluation focuses on the following scientific questions:

1. Does LEXA outperform prior work on previous benchmarks and a new challenging benchmark?
2. How does forward-looking exploration of goals compare to previous goal exploration strategies?
3. How does the distance function affect the ability to reach goals in different types of environments?
4. Can we train one general LEXA to control different robots across visually distinct environments?
5. What components of LEXA are important for performance?

We evaluate LEXA on prior benchmarks used by SkewFit [37], DISCERN [50], and Plan2Explore [43] in Section 3.3. Since these benchmarks are largely saturated, we also introduce a new challenging benchmark shown in Figure 2. We evaluate LEXA on this benchmark is Section 3.2.

### 3.1 Experimental setup

As not many prior methods have shown success on reaching diverse goals from image inputs, we perform an apples-to-apples comparison by implementing the baselines using the same world model and policy optimization as our method:

- **SkewFit** SkewFit [37] uses model-free hindsight experience replay and explores by sampling goals from the latent space of a variational autoencoder [28, 38]. Being one of the state-of-the-art agents, we use the original implementation that does not use a world model or explorer policy.
- **DDL** Dynamic Distance Learning [25] trains a temporal distance function similar to our method. Following the original algorithm, DDL uses greedy exploration and trains the distance function on the replay buffer instead of in imagination.
- **DIAYN** Diversity is All You Need [15] learns a latent skill space and uses mutual information between skills and reached states as the objective. We augment DIAYN with our explorer policy and train a learned skill predictor to obtain a skill for a given test image [12].
- **GCSL** Goal-Conditioned Supervised Learning [20] trains the goal policy on replay buffer goals and mimics the actions that previously led to the goal. We also augment GCSL with our explorer policy, as we found no learning success without it.

Our new benchmark defines goal images for a diverse set of four existing environments as follows:

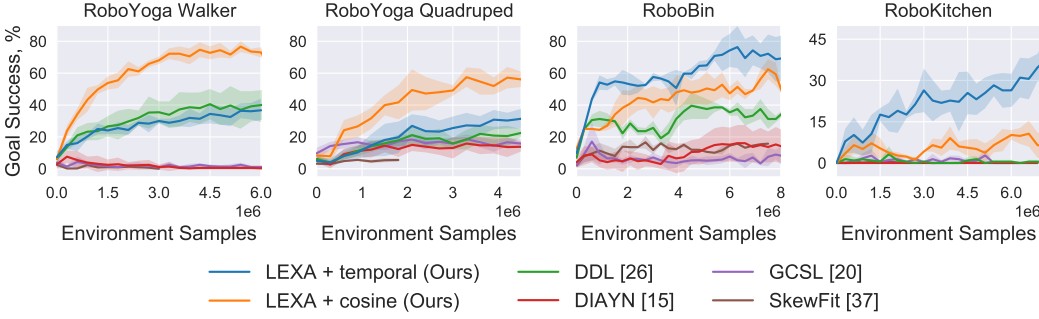

Figure 6: Evaluation of goal reaching agents on our four benchmarks. A single agent is trained from images without rewards and then evaluated on reaching goal images from the test set (see Figure 1). Both LEXA agents solve many of the tasks and significantly outperform prior work. SkewFit and DLL struggle with exploration, while DIAYN and GCSL use our explorer but still are not able to learn a good downstream policy. Refer table 1 for final success percentage (averaged across tasks) for each method and benchmark domain.

- **RoboYoga**   We use the walker and quadruped domains of the DeepMind Control Suite [48] to define the RoboYoga benchmark, consisting of 12 goal images that correspond to different body poses for each of the two environments, such as lying down, standing up, and balancing.
- **RoboBins**   Based on MetaWorld [53], we create a scene with a Sawyer robotic arm, two bins, and two blocks of different colors. The goal images specify tasks that include reaching, manipulating only one block, and manipulating both blocks.
- **RoboKitchen**   The last benchmark involves the challenging kitchen environment from [22], where a franka robot can interact with various objects including a burner, light switch, sliding cabinet, hinge cabinet, microwave, or kettle. The goal images we include describe tasks that require interacting with only one object, as well as interacting with two objects.

## 3.2   Performance on New Benchmark

We show the results on our main benchmark in Figure 6 and include heatmaps that show per-task success on each of the evaluation tasks from the benchmarks in the Appendix. Further, we report success averaged across tasks for each domain at the end of training in Table 1. We visualize example successful trajectory executions for tasks that require manipulating multiple objects in Fig. 4.

**RoboYoga**   The environments in this benchmark are directly controllable since they contain no other objects except the robot. We recall that for such settings we expect the cosine distance to be effective, as perceptual distance is quite accurate. Training is thus faster compared to using learned temporal distances, where the metric is learned from scratch. From Table 1 and Figure 6 we see that this is indeed the case for these environments (Walker and Quadruped), as LEXA with the cosine metric outperforms all prior approaches. Furthermore with temporal distances LEXA makes better progress compared to prior work on a much larger number of goals as can be seen from the per-task performance (Figures **??, ??**), even though average success over goals looks similar to that of DDL.

**RoboBins**   This environment involves interaction with block objects, and thus is not directly controllable, and so we expect LEXA to perform better with the temporal distance metric. From Table 1 and

| Method | Kitchen | RoboBins | Quadruped | Walker |
|---|---|---|---|---|
| DDL | 0.00 | 35.42 | 22.50 | 40.00 |
| DIAYN | 0.00 | 13.69 | 13.81 | 0.28 |
| GCSL | 0.00 | 7.94 | 15.83 | 1.11 |
| SkewFit | 0.23 | 15.77 | 5.52 | 0.01 |
| LEXA + Temporal (Ours) | **37.50** | **69.44** | 31.39 | 36.72 |
| LEXA + Cosine (Ours) | 6.02 | 45.83 | **56.11** | **73.06** |

Table 1: Performance on our new challenging benchmark, spanning across the four domains shown in Figure 2. The number are goal success rates, averaged over test goals within each environment.

Figure 7: Success rates on RoboBin. In line with the prior literature, previous methods are successful at reaching and sometimes pushing. LEXA pushes the state-of-the-art by picking and placing multiple objects to reach challenging goal images. Analogous heat maps for the other domains are included in the appendix.

Figure 6, we see that LEXA gets higher average success than all prior approaches. Further from the per-task performance in 7, LEXA with the temporal distance metric is the only approach that makes progress on all goals in the benchmark. The main difference in performance between using temporal and cosine distance can be seen in the tasks involving two blocks, which are the most complex tasks in this environment (the last 3 columns of the per-task plot). The best performing prior method is DDL which solves reaching, and can perform simple pushing tasks. This method performs poorly due to poor exploration, as shown in Figure 5. We see that while other prior methods make some progress on reaching, they fail on harder tasks.

**RoboKitchen** This benchmark involves diverse objects that require different manipulation behavior. From Table 1 and Figure 6 and **??** we find that LEXA with temporal distance is able to learn multiple RoboKitchen tasks, some of which require sequentially completing 2 tasks in the environment. All prior methods barely make progress due to the challenging nature of this benchmark, and furthermore using the cosine distance function makes very limited progress. The gap in performance between using the two distance functions is much larger in this environment compared to RoboBins since there are many more objects and they are not as clearly visible as the blocks.

**Single Agent Across All Environments**

In the previous sections we have shown that our approach can achieve diverse goals in different environments. However, we trained a new agent for every new environment, which doesn't scale well to large numbers of environments. Thus we investigate if we can train a train a single agent across four environments in the benchmark. From Figure **??** we see that our approach with learned temporal distance is able to make progress on tasks from RoboKitchen, RoboBins Reaching, RoboBins Pick & Place and Walker, while the best prior method on the single-environment tasks (DDL) mainly solves walker tasks and reaching from RoboBin.

### 3.3 Performance on Prior benchmarks

To further verify the results obtained on our benchmark, we evaluate LEXA on previously used benchmarks. We observe that LEXA significantly outperforms prior work on these benchmarks, and is often close to the optimal policy. Additional details are provided in **??????**.

**SkewFit Benchmark** SkewFit [37] introduces a robotic manipulation benchmark for unsupervised methods with simple tasks like planar pushing or picking. We evaluate on this benchmark in Table 2.

Table 2: Goal distance for SkewFit goals [37].

| Method | Pusher | Pickup |
|---|---|---|
| RIG [32] | 7.7cm | 3.7cm |
| RIG + HER [2] | 7.5cm | 3.5cm |
| Skew-Fit [37] | 4.9cm | 1.8cm |
| LEXA + Temporal | **2.3cm** | **1.4cm** |

Baseline results are taken from [37]. LEXA significantly outperforms prior work on these tasks. Pushing and picking up blocks from images is largely solved and future work can focus on harder benchmarks such as those introduced in our paper.

**DISCERN Benchmark** We attempted to replicate the tasks described in [50] that are based on simple two-dimensional robots [48]. While the original tasks are not released, we followed the procedure for generating the goals described in the paper. Despite following the exact procedure, we were not able to obtain similar goals to the ones used in the original paper. Nevertheless, we show the goal completion percentage results obtained with our reproduced evaluation compared to DISCERN results from the original paper. LEXA results were obtained with early stopping. In Table 3 we see that our agent solves many of the tasks in this benchmark.

**Plan2Explore Benchmark** We provide a comparison on the standard reward-based DM control tasks [48] in Table 4. To compare on this benchmark, we create goal images that correspond to the reward functions. This setup is arguably harder for our agent, but is much more practical. Note our agent never observes the reward function and only observes the goal at test time. Plan2Explore adapts to new tasks but it needs the reward function to be known at test time, while DrQV2 is an oracle agent that observes the reward at training time. Baseline results are taken from [43, 52]. LEXA results were obtained with early stopping. LEXA outperforms Plan2Explore on most tasks and even performs comparably to state of the art oracle agents (DrQ, DrQv2, Dreamer) that use true task rewards during training.

Table 3: Success for DISCERN goals [50].

| Task | LEXA | DISCERN |
|---|---|---|
| Cup | **84.0%** | 76.5% |
| Cartpole | **35.9%** | 21.3% |
| Finger | **40.9%** | 21.8% |
| Pendulum | **79.1%** | 75.7% |
| Pointmass | **83.2%** | 49.6% |
| Reacher | **100.0%** | 87.1% |

Table 4: Zero-shot return on P2E tasks [43].

| Task Zero-Shot | LEXA ✓ | P2E ✓* | DrQv2 ✗ |
|---|---|---|---|
| Walker Stand | **957** | 331 | 968 |
| Hopper Stand | **840** | **841** | 957 |
| Cartpole Balance | 886 | **950** | 989 |
| Cartpole Bal. Sparse | **996** | 860 | 983 |
| Pendulum Swing Up | **788** | **792** | 837 |
| Cup Catch | **969** | **962** | 909 |
| Reacher Hard | **937** | 66 | 970 |

### 3.4 Analysis

**Prior work** Most work we compared against struggles with exploration, such as SkewFit and DLL methods. DIAYN is augmented with our explorer, but still fails to leverage the exploration data to learn a diverse set of skills. GCSL struggles to fit the exploration data and produces behavior that does not solve the task, perhaps because the exploration data is too diverse. We observed that all baselines make progress on the simple reaching, but struggle with other tasks. We have experimented with several versions and improvements to the baselines and report the best obtained performance.

**Ablation of different components** We ablated components of LEXA on the RoboBins environment in Figure 8. Using a separate explorer policy crucial as without it the agent does not discover the more interesting tasks. Without negative sampling the agent learns slower, perhaps because the distance function doesn't produce reasonable outputs when queried on images that are more than horizon length apart. Training the distance function with real data converges to slightly lower success than using imagination data, since real data is sampled in an off-policy manner due to its limited quantity.

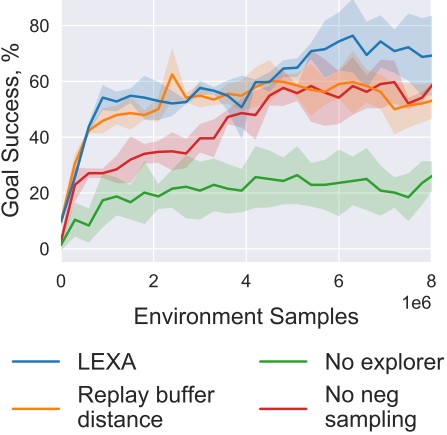

Figure 8: Ablations on RoboBins. A separate explorer is crucial for most tasks. Training temporal distance on negative samples speeds up learning, and both negative sampling and training in imagination as opposed to real data are important for the hardest tasks.

**Exploration performance** Due to importance of exploration, we further examine the diversity of the data collected during training. We log the instances where the agent coincidentally solves an evaluation task during exploration, for the RoboKitchen and RoboBins environments. In Figure 5, we see that our method encounters harder tasks involving multiple objects much more often.

# 4 Related Work

**Learning to Achieve Goals**   The problem of learning to reach many different goals has been commonly addressed with model-free methods that learn a single goal-conditioned policy [2, 26, 40]. Recent work has combined these approaches with various ways to generate training goals, such as asymmetric self-play [33, 45] or by sampling goals of intermediate difficulty [14, 18]. These approaches can achieve remarkable performance in simulated robotic domains, however, they focus on the settings where the agent can directly perceive the low-dimensional environment state.

A few works have attempted to scale these model-free methods to visual goals by using contrastive [50] or reconstructive [32, 37] representation learning. However, these approaches struggle to perform meaningful exploration as no clear reward signal is available to guide the agent toward solving interesting tasks. Some works [10, 49] avoid this challenge by using a large dataset of interesting behaviors. Other works [37, 55] attempt to explore by generating goals similar to those that have already been seen, but do not try to explore truly novel states.

A particularly relevant set of approaches used model-based methods to achieve goals via planning [13, 17] or learning model-regularized policies [35]. However, these approaches are limited by short planning horizons. In contrast, we learn long-horizon goal-conditioned value functions which allows us to solve more challenging tasks. More generally, most of the above approaches are limited by simplistic exploration, while our method leverages model imagination to search for novel states, which significantly improves exploration and in turn the downstream capabilities of the agent.

**Learning Distance Functions**   A crucial challenge for visual goal reaching is the choice of the reward or the cost function for the goal achieving policy. Several approaches use representation learning to create a distance in the feature space [9, 32, 50, 51]. However, this naive distance may not be most reflective of how hard a particular goal is to reach. One line of research has proposed using the mutual information between the current state and the goal as the distance metric [1, 12, 15, 21], however, it remains to be seen whether this approach can scale to more complex tasks.

Other works proposed temporal distances that measure the amount of time it takes to reach the goal. One approach is to learn the distance with approximate dynamic programming using Q-learning methods [16, 19, 26]. Our distance function is most similar to Hartikainen et al. [25], who learn a temporal distance with supervised learning on recent policy experience. In contrast to [25], we always train the distance on-policy in imagination, and we further integrate this achiever policy into our latent explorer achiever framework to discover novel goals for the achiever to practice on.

# 5 Conclusion

We presented Latent Explorer Achiever (LEXA), an agent for unsupervised RL that explores its environment, learns to achieve the discovered goals, and solves image-based tasks in a zero-shot way. By planning for novelty in imagination, LEXA prospectively explores to discover meaningful behaviors in substantially more diverse environments than considered by prior work. Further, LEXA is able to solve challenging downstream tasks specified as images without any supervision such as rewards or demonstrations. By proposing a challenging benchmark and the first agent to achieve meaningful performance on these tasks, we hope to stimulate future research on unsupervised agents, which we believe are fundamentally more scalable than traditional agents that require a human to design the tasks and rewards for learning.

Many challenges remain for building unsupervised agents. Many tasks in our benchmark are still unsolved and there remains room for progress on the algorithmic side both for the world model and policy optimization. Further, it is important to demonstrate the benefits of unsupervised agents on real-world systems to verify their scalability. Finally, for widespread adoption, it is crucial to consider the problem of goal specification and design methods that act on goals that are easy to specify, such as via natural language. We believe LEXA will enable future work to tackle these goals effectively.

**Acknowledgements**   We thank Ben Eysenbach, Stephen Tian, Sergey Levine, Dinesh Jayaraman, Karl Pertsch, Ed Hu and the members of GRASP lab and Pathak lab for insightful discussions. We also thank Murtaza Dalal and Chuning Zhu for help with MuJoCo environments. Finally, DP would like to thank Laura Schulz, Josh Tenenbaum and Alison Gopnik for seeding the idea of goal-setting in children, and Pulkit Agrawal for several crucial discussions (over gelato) since then. OR and KD were supported by ARL DCIST CRA W911NF-17-2-0181, ONR N00014-17-1-2093, and by Honda Research Institute. This work was partially supported by GoodAI Research Award and DARPA Machine Common Sense grant.

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
