# A Experimental Details

**Environments** The episode length is 150 for RoboBin and RoboKitchen and 1000 for RoboYoga. We show all goals in Figure A.1. For both *Walker* and *Quadruped*, the success criterion is based on the largest violation across all joints. The global rotation of the Quadruped is expressed as the three independent Euler angles. Global position is not taken into account for the success computation. *RoboBin*. The success criterion is based on placing all objects in the correct position within 10 cm. For reaching task, the success is based on placing the arm in the correct position within 10 cm. *RoboKitchen* uses 6 degrees of freedom end effector control implemented with simulation-based inverse kinematics. The success criterion is based on placing all objects in the correct position with a threshold manually determined by visual inspection. Note that this is a strict criterion: the robot needs to place the object in the correct position, while not perturbing any other objects.

**Evaluation** We reported success percentage at the final step of the episode. All experiments on our benchmark as well as on the SkewFit benchmark were ran 3 seeds. Due to large required compute, DISCERN and Plan2Explore results for LEXA were only run with one seed. The DISCERN and Plan2Explore results should therefore not be used for rigorous comparisons, but are nevertheless indicative of the simplicity of these benchmarks. Plots were produced by binning every 3e5 samples. Heatmap shows performance at the best timestep. Each model was trained on a single high-end GPU provided by either an internal cluster or a cloud provider. The training took 2 to 5 days. The final experiments required approximately 100 training runs, totalling approximately 200 GPU-days of used resources.

**Implementation** We base our agent on the Dreamer implementation. For sampling goals to train the achiever, we sample a batch of replay buffer trajectories and sample both the initial and the goal state from the same batch, therefore creating a mix of easy and hard goals. To collect data in the real environment with the achiever, we sample the goal uniformly from the replay buffer. We include code in the supplementary material. The code to reproduce all experiments will be made public upon the paper release under an open license.

**Hyperparameters** LEXA hyperparameters follow Dreamer V2 hyperparameters for DM control (which we use for all our environments). For the explorer, we use the default hyperparameters from the Dreamer V2 codebase [25]. We use action repeat of 2 following Dreamer. LEXA includes only one additional hyperparameter, the proportion of negative sampled goals for training the distance function. It is specified in Table A.5. The hyperparameters were chosen by manual tuning due to limited compute resources. The base hyperparameters are shared across all methods for fairness.

**DIAYN baseline** We found that this baseline performs best when the reverse predictor is conditioned on the single image embedding $e$ rather than latent state $s$. We use a skill space dimension of 16 with uniform prior and Gaussian reverse predictor with constant variance. For training, we produce the embedding using the embedding prediction network from Section 2.4. We observed that DIAYN can successfully achieve simple reaching goals using the skill obtained by running the reverse predictor on the goal image. However, it struggles with more complex tasks such as pushing, where it only matches the robot arm.

**GCSL baseline** We found that this baseline performs best when the policy is conditioned on the single image embedding $e$ rather than latent state $s$. This baseline is trained on the replay buffer images and only uses imagined rollouts to train an explorer policy. For training, we sample a random image from a trajectory and sample the goal image from the uniform distribution over the images later in the trajectory following [20]. We similarly observe that this baseline can perform simple reaching goals, but struggles with more complex goals.

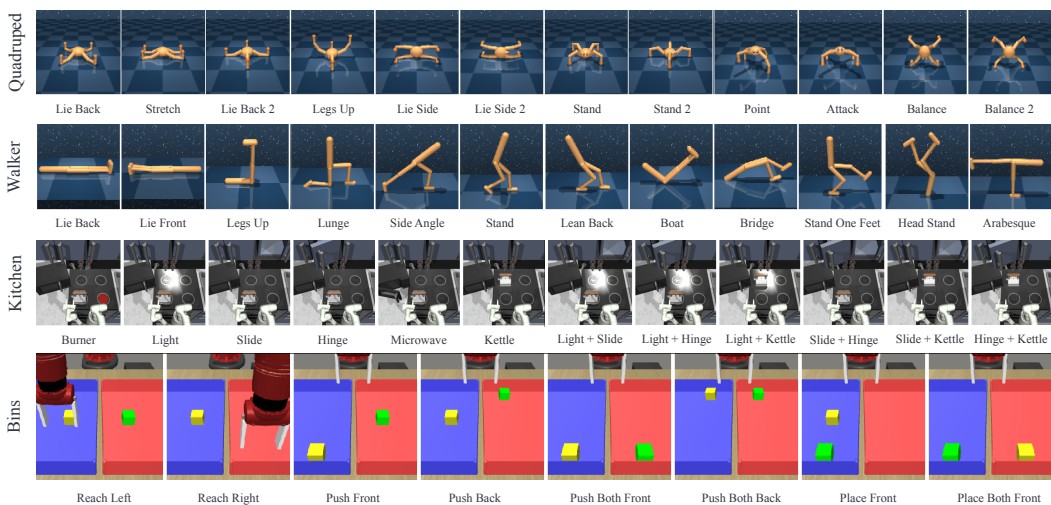

Figure A.1: All goals for the four environments that we consider. Our benchmark further includes an additional set of even harder goals, available in the repository.

| Algorithm | From Pixels | Zero-Shot | Exploration | Planning |
|---|---|---|---|---|
| CTS [4], Curiosity [35], RND [8] | ✓ | ✗ | ✓ | ✗ |
| Plan2Explore [44] | ✓ | ✗ | ✓ | ✓ |
| HER [2] | ✗ | ✓ | ✗ | ✗ |
| Visual Foresight [13] | ✓ | ✓ | ✗ | ✓ |
| Actionable Models [10] | ✓ | ✓ | ✗ | ✗ |
| DIAYN [15] | ✗ | ✓ | ✓ | ✗ |
| Asymmetric Self-Play [34] | ✗ | ✓ | ✓ | ✗ |
| SkewFit [38] | ✓ | ✓ | ✓ | ✗ |
| Go-Explore [14] | ✓ | ✓ | ✓ | ✗ |
| LEXA (Ours) | ✓ | ✓ | ✓ | ✓ |

Table A.1: Conceptual comparison of unsupervised reinforcement learning methods. LEXA combines forward-looking exploration by planning with achieving downstream tasks zero-shot while learning purely from pixels without any privileged information.

Table A.5: Hyperparameters for LEXA over the Dreamer default hyperparameters.

| Hyperparameter | Value | Considered values |
|---|---|---|
| Action repeat (all environments) | 2 | 2 |
| Proportion of negative samples | 0.1 | 0, 0.1, 0.5, 1 |
| Proportion of explorer:achiever data collected in real environment | 1:1 | 1:1 |
| Proportion of explorer:achiever training imagination rollouts | 1:1 | 1:1 |

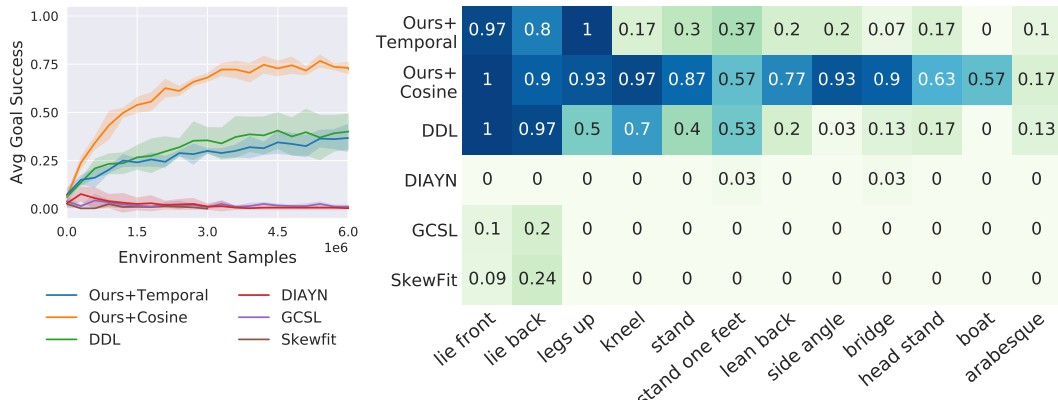

Figure A.2: **RoboYoga Walker Benchmark.** Left: success rates averaged across all 12 tasks. Right: final performance on each specific task, ranging from light green (0) to dark blue (100%). We observe that the simple latent cosine distance function works well on this task, substantially outperforming other competing agents. In the heatmap, most agents can solve the easy tasks, but only LEXA makes progress on solving a majority of the tasks and achieves good performance.

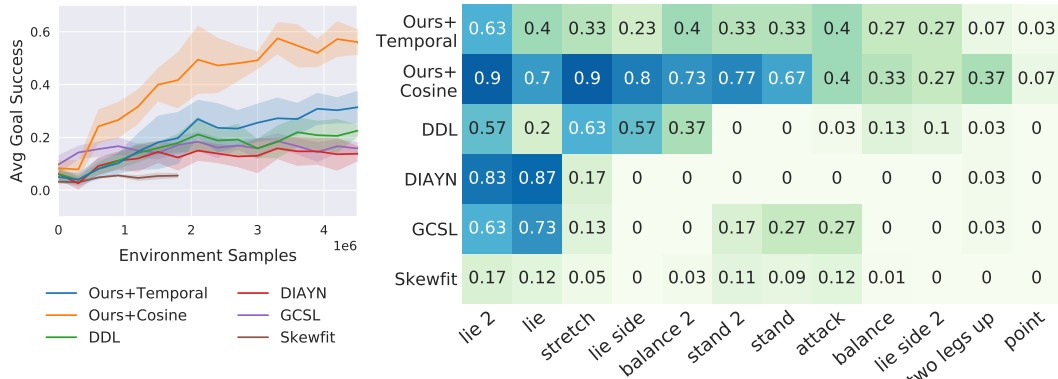

Figure A.3: **RoboYoga Quadruped Benchmark.** Left: success rates averaged across all 12 tasks. Right: final performance on each specific task, ranging from light green (0) to dark blue (100%). We observe that the simple latent cosine distance function works well on this task, substantially outperforming other competing agents. In the heatmap, most agents can solve the easy tasks, but only LEXA makes progress on solving a majority of the tasks and achieves good performance.

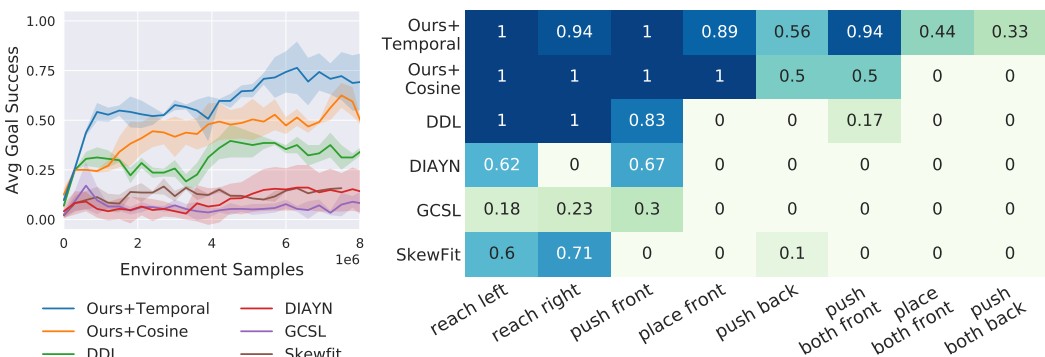

Figure A.4: **RoboBin Benchmark.** Left: success rates averaged across all 8 tasks. Right: final performance on each specific task. While cosine distance works on simple goals, temporal distance outperforms it on tasks requiring manipulating several blocks (last three columns), as this distance focuses on the part of the environment that's hardest to manipulate. Prior agents only solve the easiest reaching tasks, struggling either with exploration or learning the downstream policy.

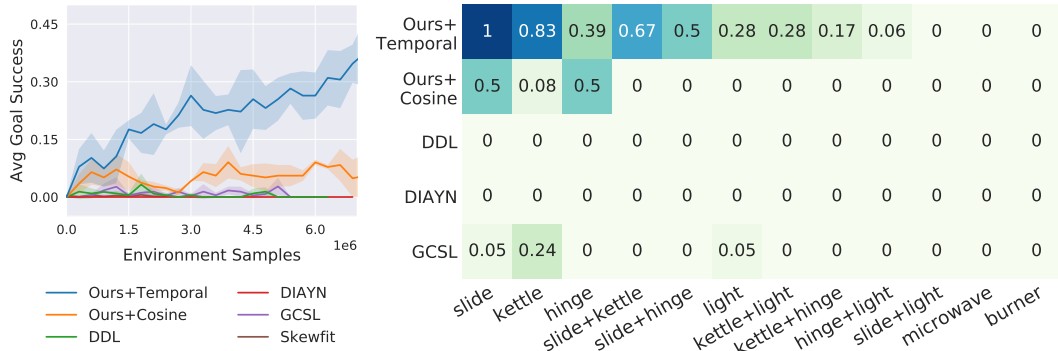

Figure A.5: **RoboKitchen Benchmark.** Left: success rates averaged across all 12 tasks. Right: final performance on each specific task. RoboKitchen is challenging both for exploration and downstream control, with most prior agents failing all tasks. In contrast, LEXA is able to learn both an effective explorer and achiever policy. Temporal distance helps LEXA focus on small parts such as the light switch, necessary to solve these tasks. LEXA makes progress on four out of six base tasks, and is even able to solve combined goal images requiring e.g. both moving the kettle and opening a cabinet.

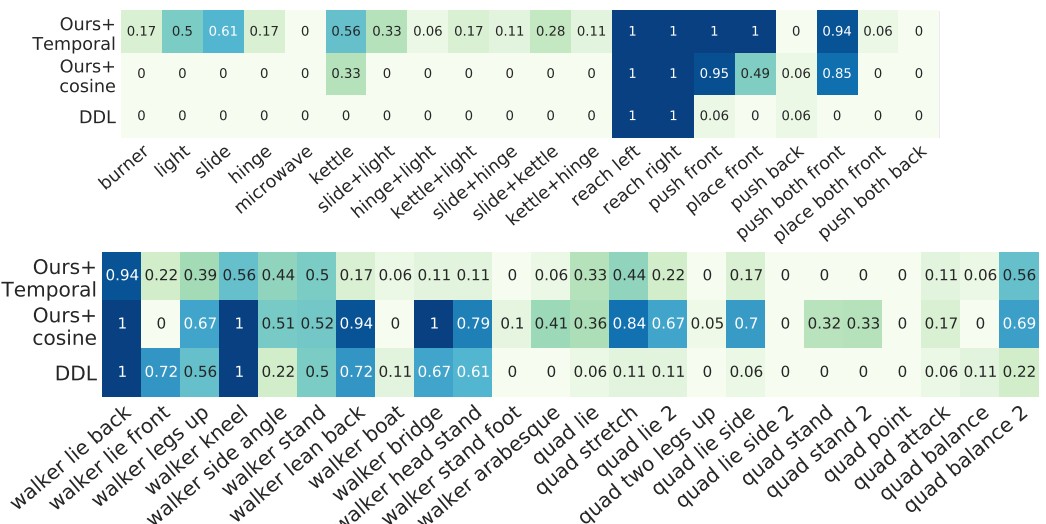

Figure A.6: **Single agent** trained across Kitchen, RoboBin, Walker, with final performance on each specific task. LEXA with temporal distance is able to make progress on tasks from all environments, while LEXA+cosine and DDL don't make progress on the kitchen tasks.

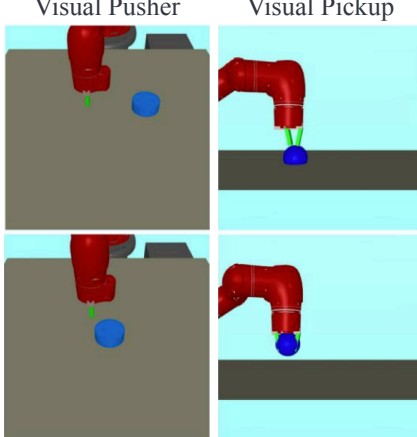

Visual Pusher    Visual Pickup

Table A.2: Results on SkewFit tasks [38].

| Method | Visual Pusher | Visual Pickup |
|---|---|---|
| LEXA + temporal | **0.023** | **0.014** |
| Skew-Fit [38] | 0.049 | 0.018 |
| RIG [33] | 0.077 | 0.037 |
| RIG + Hazan et al. | 0.059 | 0.039 |
| RIG + HER [2] | 0.075 | 0.035 |
| DISCERN [51] | 0.094 | 0.039 |
| RIG + Goal GAN [18] | 0.088 | 0.039 |
| RIG + DISCERN-g | 0.07 | 0.032 |
| RIG + # Exploration | 0.088 | 0.04 |
| RIG + Rank-Based | 0.067 | 0.035 |

Figure A.7: Final goal reaching error in meters on tasks from SkewFit [38]. Example observations are provided on the left. Baseline results are taken from [38]. LEXA significantly outperforms prior work on these tasks. Pushing and picking up blocks from visual observations is largely solved, so future work will likely focus on harder benchmarks such as the one proposed in our paper.

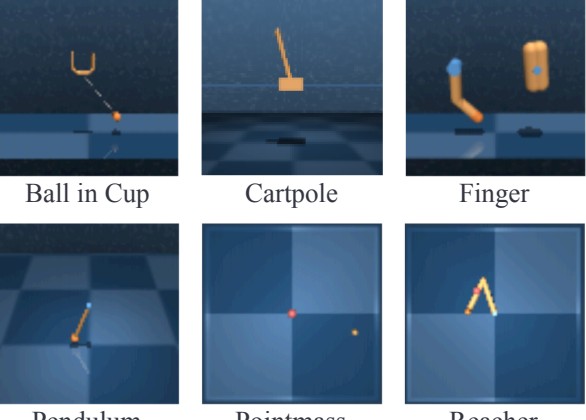

Ball in Cup    Cartpole    Finger

Pendulum    Pointmass    Reacher

Table A.3: Results on DISCERN tasks* [51].

| | LEXA | DISCERN* [51] |
|---|---|---|
| Ball in cup | **84%** | 76.5% |
| Cartpole | **35.9%** | 21.3% |
| Finger | **40.9%** | 21.8% |
| Pendulum | **79.1%** | 75.7% |
| Pointmass | **83.2%** | 49.6% |
| Reacher | **100%** | 87.1% |

Figure A.8: Goal success rate on the tasks replicated from [51]. Example observations are provided on the left. LEXA results were obtained with early stopping. *While the original tasks are not released, we followed the procedure for generating the goals described in [51]. Despite following the exact procedure, we were not able to obtain similar goals to the ones used in the original paper. Nevertheless, we show the goal completion percentage results obtained with our reproduced evaluation compared to DISCERN results from the original paper. We see that our agent solves many of the tasks in this benchmark and performs better on this comparison. We further suspect that the goals that we generated are harder than the ones used in the original paper, such as in the cartpole environment where our goals require swinging the pole higher up. Future work will likely focus on harder benchmarks such as our RoboYoga benchmark, rather than these simple robots with one or two degrees of freedom.

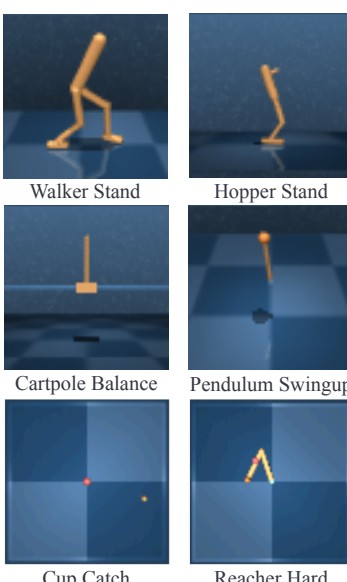

Walker Stand

Hopper Stand

Cartpole Balance

Pendulum Swingup

Cup Catch

Reacher Hard

Table A.4: Results on zero-shot DeepMind control tasks from Plan2Explore [44].

| Method | LEXA | P2E [44] | DrQv2 [53] |
|---|---|---|---|
| Zero-Shot | ✓ | ✓* | ✗ |
| Walker Stand | **957** | 331 | 968 |
| Hopper Stand | **840** | **841** | 957 |
| Cartpole Balance | 886 | **950** | 989 |
| Cartpole Balance Sparse | **996** | 860 | 983 |
| Pendulum Swingup | **788** | **792** | 837 |
| Cup Catch | **969** | **962** | 909 |
| Reacher Hard | **937** | 66 | 970 |

Figure A.9: Final return on DM control tasks [49]. Example goals achieved by LEXA are provided to the left. Baseline results taken from [44, 53]. LEXA results were obtained with early stopping. *Plan2Explore adapts to new tasks but it needs the reward function to be known at test time while LEXA does not require access to rewards. To compare on the same benchmark, we create goal images that correspond to the reward functions. This setup is arguably harder for our agent, but is much more practical. Our agent outperforms Plan2Explore on most tasks and even performs comparably to state of the art oracle agents (DrQ, DrQv2, Dreamer) that use true task rewards during training.