# OpenReview forum: "Discovering and Achieving Goals via World Models"
_NeurIPS.cc/2021/Conference — NeurIPS 2021 Poster_

### Official Review · Reviewer_LRem · 2021-07-10

**Rating:** 6
**Confidence:** 4

**Summary:**

This paper introduces a framework to train policies for visual goal oriented tasks. The framework comprises two policies, the first, an explorer, which is trained to seek out novel states (where novelty is defined as the level of agreement between an ensemble of future state predictors - an approximate measure of epistemic uncertainty), while the second, an achiever, is trained to reach these states. A model based approach to train the achiever, where the achiever is trained using trajectories generated by a learned dynamics model. The paper also introduces a new benchmark for testing goal oriented polices of this form, comprising a kitchen manipulation task, a block handling task, and a yoga task.

**Ethical Concerns:**

I have no ethical concerns about this work.

**Limitations And Societal Impact:**

I don't see immediate negative societal impact for this work, as it is extremely early stage research.

However, a number of limitations were not discussed, and should be addressed in the paper.

A major challenge preventing the deployment and practicality of exploratory policies in real world settings is safety. Looking at the motions followed by the policy, and given the ease with which torque controlled robots like the panda can be broken, it is clear that these policies are completely infeasible for real-world robots. I think it is important to make this clear, and reflect on the gaps between sim and real here, and the need for stringent safety constraints if this was attempted in real world settings.

Moreover, the proposed approach will rely significantly on environment resetting, which is also infeasible in real world settings. Additional detail about the number of times and conditions under which these environments were reset, episode length etc. and being explicit about this would be valuable.

**Main Review:**

Originality:

The primary contribution of this paper is to show that an architecture explicitly balancing exploration and exploitation is also effective in high dimensional settings for goal oriented robot tasks. Existing work has focused on lower dimensional observations. I like the idea, and think its a good one, although the originality seems to be limited to scaling existing ideas to harder tasks.

The second contribution of the paper is a new suite of goal oriented tasks for robot goal tasks. These tasks seem to be more complex (in terms of manipulation, mapping of actions to goals) than existing benchmark suites, although I think some additional discussion on this may be warranted, eg.  benchmarks like robothor https://ai2thor.allenai.org/robothor/ are already widely used for goal oriented visual navigation, and have physical counterparts that are better suited for real world transfer.

Quality and clarity:

The idea is solid, and experiments are clear and concise. Ablations highlight the importance of the explorer network. A number of baselines and comparisons to prior work are included.

Significance:

The results presented here are unsurprising, but solid. I think the benchmark of tasks is a nice contribution, and that the  demonstration that policy training balancing exploration and exploitation is effective in high dimensions is a useful.

Detailed questions/comments:

Reward engineering. The introduction notes that a challenge with RL lies in reward engineering, where it can be difficult to choose effective rewards for tasks, and hence the need for unsupervised goal oriented learning. While I agree with this in principle, the results in this paper seem to me to contradict this, as it is clear that different cost functions eg. the cosine vs temporal distance measure, are still needed to train policies for different tasks. It seems to me that all that has happened here is that reward engineering has shifted to a different latent space.

Could you clarify how the explorer and achiever were trained. Is this jointly, or is the explorer trained first, with the achiever then trained, by bootstrapping on data gathered by the explorer? From the text I gather the latter, but I would appreciate a bit more clarity here, eg. how much data was seen by each, etc.

Results: How is success determined for each of the tasks? Is this distance to a particular simulator state? A perceptual visual metric? Some human label?

In some of the example videos, it seems that camera perspective effects is affecting the type of goals reached. How robust is this sort of approach to small changes in camera angle, etc.

Minor:

Typos: line 66, the introduce the
line 129-230, grammar


**Time Spent Reviewing:**

1.5

---

> ### Author Response · Authors · 2021-08-11
> **Author Response: provided significant new experiments on prior benchmarks**
>
> Thanks for the positive response and constructive feedback!
> We would like to point the reviewer to the significant additional results requested by other reviewers in the general response.  We believe that our paper constitutes a significant advance in goal-conditioned learning by being able to solve much harder tasks considered in our paper (see the details on the supplementary website: https://sites.google.com/view/exanet/home) , while performing well on simpler tasks considered by prior work.
>
>
> > “The results presented here are unsurprising, but solid.”
> - We would like to emphasize that the environments we consider are significantly more challenging than those considered by previous work in unsupervised goal discovery. The kitchen environment is especially challenging with 6 separate objects. Prior robot learning based approaches relied on expert demonstrations [1] to solve these tasks, whereas our approach makes significant progress without any supervision (demonstrations nor rewards).
>
> - With the roboBins environment our approach is able to pick and place two different objects in sequence, whereas the closest related environment from prior work in the unsupervised learning literature (Skewfit pickup environment) constrained arm motion to the plane containing a single object, and didn’t require placing the object.
>
> - Our locomotion environments further feature high-dimensional agents like the quadruped, whereas prior work has only shown success on simple 2-dimensional environments. We therefore believe that our approach constitutes a significant advance over prior work.
>
> > “Could you clarify how the explorer and achiever were trained.Is this jointly, or is the explorer trained first, with the achiever then trained, by bootstrapping on data gathered by the explorer?”
> - The explorer and achiever are trained jointly, indicated by lines 4 and 5 in the algorithm box.
>
> > “How is success determined for each of the tasks?”
> - The success is determined based on distance to a particular simulator state, as specified in the supplementary material. Please see Appendix A, under environments for more details.
>
> > "the originality seems to be limited to scaling existing ideas to harder tasks"
> - We emphasize that our major contribution is to investigate the importance of exploration for goal-conditioned learning (GCRL). Different from prior work in GCRL, our agent does not simply revisit goals from the replay buffer, but attempts to discover novel goals through multi-step imagination. Our empirical results (Fig.8 in the paper, Fig.1 in the Appendix) show that such forward-looking exploration is crucial for good performance in GCRL, our paper being both the first to investigate forward-looking exploration for GCRL and the first to empirically achieve good exploration on complex environments such as kitchen manipulation from pixels.
>
> > “additional discussion ..eg. benchmarks like robothor https://ai2thor.allenai.org/robothor/ are already widely used for goal oriented visual navigation, and have physical counterparts that are better suited for real world transfer”
> - We agree that the robothor environment is promising for evaluating long-term planning agents, and the ability to solve very diverse tasks. In our paper, we focus on challenging control tasks such as manipulation and high-dimensional locomotion, instead of visual navigation where control is comparatively simple. However, we believe that mobile manipulation is an important avenue for future work on unsupervised agents and we will add this to the future work discussion.
>
> > “different cost functions eg. the cosine vs temporal distance measure, are still needed ...reward engineering has shifted to a different latent space."
>
> - While we do have two different cost functions, we use cosine distance for tasks that only require pose matching of the robot since this corresponds to ‘exactly matching’ the goal image.
> On the other hand, the temporal distance is used for all tasks involving manipulation of objects.
> The temporal distance uses timesteps it takes to move from one image to another as a distance measure. This ignores large changes in robot position, since these can be completed in very few steps, and will instead focus more on the objects.
>
> - This is a far more principled approach to using a cost function than reward engineering in RL, where for each new environment developers need to design a task dependent reward function. Reward engineering typically requires considering each object and each stage of the task separately (go to the object, pick up object, move to different bin, etc..), while our approach can learn even without explicitly providing these stages.
>
> > “..camera perspective effects is affecting the type of goals reached. How robust is this sort of approach to small changes in camera angle, etc.”
> - Since our agent only takes in visual observations, it might be harder to solve tasks where the objects are further from the camera. This is evidenced for example on the RoboBin domain, where pushing blocks to the front is easier than pushing them to the back. This is a natural phenomenon in visual robot learning. However, in preliminary experiments, we found that our agent is robust to different camera angles and in particular does not require the camera to be aligned with any of the environment axes. We have observed that several different camera angles all work comparatively well for our tasks.
>
> >  “Looking at the motions followed by the policy, and given the ease with which torque controlled robots like the panda can be broken, it is clear that these policies are completely infeasible for real-world robots...reflect on the gaps between sim and real here, and the need for stringent safety constraints if this was attempted in real world settings.”
>
> -  We agree that making our policies safe for real world deployment is a major direction for future work. The robotic agents in the paper all use end effector position control. With a slower execution speed, similar policies should be feasible to be learned and deployed on robots. We do agree that there are gaps between sim and real in terms of safety considerations, since the robot could have unsafe collisions. Some steps that could improve real world safety include padding the links of the robot and constraining the reachable robot state space to avoid collisions with large, immovable obstacles (eg : fridge, oven etc ). We will add this discussion in the paper.
>
> > “..will rely significantly on environment resetting, which is also infeasible in real world settings. Additional detail about the number of times and conditions under which these environments were reset, episode length etc. and being explicit about this would be valuable.”
>
> - We refer the reviewer to Appendix 2 about specific hyperparameters. The experiments in our paper used regular episode resets, as is common in robot learning work. However, we agree that this is in general an unreasonable assumption for real world learning, and relaxing this assumption is a promising direction of future work. We further believe that our agent is particularly suited for the reset-free setup since a reset is just a different task for the agent to autonomously discover. For example, instead of just learning to open cabinets, the robot will also need to learn how to close the cabinet.
>
>
> [1] : Relay Policy Learning: Solving Long-Horizon Tasks via Imitation and Reinforcement Learning (Gupta et al)

---

### Official Review · Reviewer_dFJ7 · 2021-07-16

**Rating:** 6
**Confidence:** 4

**Summary:**

This paper proposes a model based RL setting where an agent learns a task agnostic model, learns two policies to explore and reach input goal states (specified as images) respectively and is able to operate in a goal conditional mode during evaluation time.

The key thesis is to use the world model to incentivize the agent to explore the environment by maximizing expected information gain and reach input goal states by learning to reach states from its past experiences.

There have been other model free and model based approaches studying the same problem but the claim is that its the first to combine model based exploration and reachability on new benchmark environments.

**Limitations And Societal Impact:**

Seems adequate

**Main Review:**

Unsupervised RL is an interesting and important problem in deep and reinforcement learning. There have been several papers and baselines that have tackled this problem before. Perhaps the biggest limitation of this paper is lack of comparisons to past benchmarks. For instance the Sekar et al. paper (cited in the main paper) considers several environments and tasks that are not covered in this paper. The DISCERN (cited) model also provided benchmark results on Atari and continuous control tasks. The benchmark in this paper is new and therefore it is difficult to understand and compare the approach with past results.

The main argument that the authors present regarding past benchmarks is -- "In contrast to common RL benchmarks such as Atari [5] that require training over 50 different agents and thus enormous computational resources, our unsupervised RL benchmark only requires training 4 agents, which are then evaluated across many tasks, allowing for faster iteration time and making the research more accessible". It seems practical to at least try a handful of the same environments to set the right experimental standards. Otherwise it is very difficult to validate the claims and understand the effect of choices proposed in this paper. I believe the paper would become much stronger if this is addressed.

**Time Spent Reviewing:**

2

---

> ### Author Response · Authors · 2021-08-11
> **Author Response: provided significant new experiments on prior benchmarks**
>
> Thank you for the helpful and insightful feedback! In our original submission, we evaluated our method on a new benchmark designed to be much harder than tasks considered in prior work (see the details on the supplementary website: https://sites.google.com/view/exanet/home). However, we agree that evaluation on prior benchmarks is helpful to put our results in context, and we have now additionally evaluated on 15 new environments from prior work as outlined below. Our agent performs comparably or better than all previous methods on these environments, without us changing any of its hyperparameters. We would like to ask the reviewer to let us know should there remain any concerns that prevent you from accepting the paper.
>
> > “Sekar et al. paper (cited in the main paper) considers several environments and tasks that are not covered in this paper”
>
> Plan2Explore evaluates on standard reward-based tasks, while our agent observes no rewards. However, we were able to adapt some of the tasks into the goal-based setup by providing appropriate goals to our agent. Note that this setup is arguably harder for our agent, since while Plan2Explore assumes the availability of the reward at test time (but not during exploration, which is entirely unsupervised), our agent needs to learn the reward just based on the goal image. Our setup is further more practical than assuming availability of the reward since only a goal image needs to be provided. Despite being in this harder setup, our agent outperforms Plan2Explore on most tasks, and even performs comparably to state of the art oracle agents that explore with reward signal.
>
> |                       | DrQ | DrQv2 | Dreamer | Plan2Explore | Ours |
> |-------------------------|-----|-------|---------|-----|------|
> | Zero-shot               | ❌   | ❌     | ❌       | ✅   | ✅    |
> | Walker stand            | 957 |   968 |     977 | 331 | **957**  |
> | Hopper stand            | 930 |   957 |     923 | **841** | **840**  |
> | cartpole balance        | 973 |   989 |     979 | **950** | 886  |
> | cartpole balance sparse | 983 |   983 |     941 | 860 | **996**  |
> | pendulum swingup        | 383 |   837 |     833 | **792** | **788**  |
> | cup catch               | 962 |   909 |     962 | **962** | **969**  |
> | reacher hard            | 392 |   970 |     817 |  66 | **937**  |
>
> > “The DISCERN (cited) model also provided benchmark results”
>
>  We have attempted to produce an experimental comparison on the DISCERN benchmarks. The benchmark goals and evaluation procedure are not released, and the authors didn’t respond to us about replication in time for the rebuttal deadline. Therefore, we followed the goal generation procedure as described in the DISCERN paper. Despite following the exact procedure, we were not able to generate the same goals as in the original paper, for instance, on the cartpole benchmark the original paper shows goals that have the pole in diverse positions, while we obtained goals with the pole only facing upwards. We believe the authors could have been using a different internal version of the DMC suite, or perhaps used an unspecified action repeat. Nevertheless, we show the results obtained with our regenerated goals compared to DISCERN results from the original paper below. We see that our agent performs adequately on this benchmark and appears to be competitive. We further suspect that the goals we generated are harder than the ones used in the original paper, such as in the cartpole environment (our upwards-facing goals are strictly harder because they require careful balance).
>
> |             | DISCERN  | Ours     |
> |-------------|----------|----------|
> | ball in cup | 76.5% | **84%**       |
> | cartpole    | 21.3%     | **35.9%** |
> | finger      | 21.8%     | **40.9%** |
> | pendulum    | 75.7%     | **79.1%** |
> | pointmass   | 49.6% | **83.2%**   |
> | reacher     | 87.1% | **100%**     |
>
>
> > “The benchmark in this paper is new and therefore it is difficult to understand and compare the approach with past results.”
>
>  We now provided experimental results on prior benchmarks above as well as on the SkewFit benchmark. Our agent significantly outperforms prior work on the SkewFit tasks.
>
> |               | RIG   | RIG + Hazan et al. | RIG + HER | DISCERN | RIG + AutoGoal GAN | RIG + DISCERN-g | RIG + # Exploration | RIG + Rank-Based Priority | Skew-Fit | Ours  |
> |---------------|-------|--------------------|-----------|---------|--------------------|-----------------|---------------------|---------------------------|----------|-------|
> | Visual Pusher | 0.077 |              0.059 |     0.075 |   0.094 |              0.088 |            0.07 |               0.088 |                     0.067 |    0.049 | **0.023** |
> | Visual Pickup | 0.037 |              0.039 |     0.035 |   0.039 |              0.039 |           0.032 |                0.04 |                     0.035 |    0.018 | **0.014** |
>
>
> We stress that these prior benchmarks are no longer appropriate for evaluation since they were largely saturated by prior work, which motivated our new proposed benchmark. We believe our benchmark is an important part of our contribution. The experimental results on prior benchmarks that we now provide further confirm this.

---

> ### Author Response · Authors · 2021-08-17
> **Discussion Period**
>
> Dear Reviewer,
>
> Hope you got a chance to read our rebuttal. Please let us know whether you have any further concerns remaining that prevent you from accepting the paper.

---

> > ### Comment · Reviewer_dFJ7 · 2021-08-23
> > **Post rebuttal**
> >
> > It is great to see the authors trying their methods on more standard environments. I also think that the method is promising but I do not believe the experimental setup is water tight. For instance, the authors acknowledge that "on the cartpole benchmark the original paper shows goals that have the pole in diverse positions, while we obtained goals with the pole only facing upwards". I think the authors need to do another round of more careful experimental validation before it is ready for publication.

---

> > > ### Author Response · Authors · 2021-08-23
> > > **Authors Response**
> > >
> > > Dear Reviewer,
> > >
> > > Thank you very much for engaging in the discussion! We understand that you are concerned about whether *"the experimental setup is water tight"*. The example you pose is our replication of the DISCERN benchmark, that you requested in your review. Unfortunately, the DISCERN paper does not provide sufficient detail to replicate their goal images precisely, and their original goal images were not released publicly. The authors were not responsive when contacted via email. Thus, we have replicated the tasks as accurately as possible given the publicly available information, as we describe in our first comment.
> > >
> > > You wrote *"another round of more careful experimental validation before it is ready for publication"* but this suggestion is vague and not actionable for us. Do we understand correctly that you have additional concerns about our experimental setup that go beyond the DISCERN benchmark? If so, could you please clarify exactly what the issues are? We are confident in our experimental setup for all other experiments (RoboKitchen, RoboBins, RoboYoga, SkewFit Pickup, SkewFit Pushing, and the 7 Plan2Explore environments) and will be happy to address any of your concerns.
> > >
> > > Thank you!

---

> > > > ### Comment · Reviewer_dFJ7 · 2021-08-25
> > > > **response**
> > > >
> > > > I am generally satisfied with the experiments you have done and will change my scores to reflect this. But I am not sure why it is precisely difficult to replicate the control suite setup exactly as in DISCERN. For instance in the cart pole example, the environment was reset to the balance set which can ensure diverse states under random exploration. These might be minor things but it raises concerns for me about the experimental soundness.

---

> > > > > ### Author Response · Authors · 2021-08-26
> > > > > **Authors Response**
> > > > >
> > > > > > I am generally satisfied with the experiments you have done and will change my scores to reflect this.
> > > > >
> > > > > Thank you for planning to update your score!
> > > > >
> > > > > > But I am not sure why it is precisely difficult to replicate the control suite setup exactly as in DISCERN. For instance in the cart pole example, the environment was reset to the balance set which can ensure diverse states under random exploration.
> > > > >
> > > > > We closely followed the instructions in the DISCERN paper:
> > > > >
> > > > > *"The goals are generated by acting randomly for 25 environment steps after initialization. In the case of cartpole, we draw the goals from a random policy acting in the environment set to the balance task, where the pole is initialized upwards in order to generate a more diverse set of goals against which to measure"*
> > > > >
> > > > > However, the upright position of the balance task and the diverse goals are more than 25 steps away, so it is not possible to generate them by acting randomly for 25 steps as the paper suggests. We conclude that the authors of the DISCERN paper either (1) used more than 25 random actions for cartpole, (2) used an unspecified action repeat, or (3) used an earlier version of the Control Suite that is internal to DeepMind and differs from the publicly released environments. We contacted the authors regarding this but did not receive a response and their code is not publicly available either. Thus, it is not possible to reproduce the DISCERN benchmark exactly, which we highlighted in our initial response.
> > > > >
> > > > > > These might be minor things but it raises concerns for me about the experimental soundness.
> > > > >
> > > > > Do we understand correctly that the experimental soundness of experiments other than the DISCERN benchmark is still a concern for you? If so, please point to the specific issues. We are confident in our experimental results and will be happy to address any remaining concerns.

---

> > > > > > ### Author Response · Authors · 2021-08-31
> > > > > > **Request for follow up**
> > > > > >
> > > > > > Dear Reviewer,
> > > > > >
> > > > > > We were wondering if you had an opportunity to look at our latest response. In your earlier reply, you mentioned that you plan to change the score. We wanted to send a gentle reminder about that since the deadline is just two days away. Please let us know if there are more questions.
> > > > > >
> > > > > > Thanks a lot for engaging in the discussion!

---

### Official Review · Reviewer_uhSG · 2021-07-16

**Rating:** 6
**Confidence:** 3

**Summary:**

Summary: This paper proposes a model-based method with an ensemble disagreement-based exploration bonus for learning universal goal-conditioned policies. The world model and policy learning methods are adapted from DREAMER. In addition to that, the paper first introduces an explorer to first explore the state with large uncertainties, and then train the achiever to reach the proposed goals. The use of the world model enables data-efficient learning and exploration. The paper makes solid experiments and evaluates their approach on various domains. The paper also discusses two different distance metrics for learning goal-conditioned policy on images.

**Limitations And Societal Impact:**

The authors claim that "Potential societal impacts of robot learning are at this stage largely hypothetical since the technology is not yet in widespread production." and I agree.

**Main Review:**

- The paper is well-written with convincing experiments. The proposed approach is simple yet effective. The separation of the exploration policy for goal generation and goal-conditioned policy learning looks like a general approach. The experimental results are not perfect. I notice that most tasks can't reach a 100 percent success rate. But considering the challenging image-input setup, it's great to see that it works in practice.
- What concerns me most is the novelty of the paper. The paper's technique is a combination of image-based RL, uncertainty-based exploration, and goal-generation tasks, but I am still happy to see that one can assemble these techniques to solve challenging problems. I would be happy to see it at the Neurips conference.

**Time Spent Reviewing:**

3

---

> ### Author Response · Authors · 2021-08-11
> **Author Response: provided significant new experiments on prior benchmarks**
>
> Thank you for the positive response and constructive feedback! We would also like to point the reviewer to the significant additional results requested by other reviewers in the general response. We believe that our paper constitutes a significant advance in goal-conditioned learning by being able to solve much harder tasks considered in our paper (see the details on the supplementary website: https://sites.google.com/view/exanet/home), while performing well on simpler tasks considered by prior work.
>
> > “I notice that most tasks can't reach a 100 percent success rate.”
>
> Note that from the 40 tasks we evaluated, our method solves 19 tasks with high success rate (more than 75%), and many goals are indeed solved to close to 100% success rate. Because our agent needs to solve all possible goals within a single policy, we hypothesize it is hard for each individual goal to be solved perfectly. This is in contrast to standard RL benchmarks where it can be easy to overfit to a particular goal, but, on the contrary, perhaps somewhat alike human performance, where humans are able to solve many different tasks but sometimes imperfectly. Further, we include very challenging tasks on which even our method fails in our benchmark to stimulate further research on unsupervised goal reaching.
>
> > “What concerns me most is the novelty of the paper.”
>
> We emphasize that our major contribution is to investigate the importance of exploration for goal-conditioned learning (GCRL). Different from prior work in GCRL, our agent does not simply revisit goals from the replay buffer, but attempts to discover novel goals through multi-step imagination. Our empirical results further show that such forward-looking exploration is crucial for good performance in GCRL, our paper being both the first to investigate forward-looking exploration for GCRL and the first to empirically achieve good exploration on complex environments such as kitchen manipulation.

---

### Official Review · Reviewer_bDZB · 2021-07-16

**Rating:** 6
**Confidence:** 3

**Summary:**

This paper proposes to jointly train two agents in a differentiable world model: an *explorer* that is rewarded to explore uncertain states and an *achiever* whose task is to achieve a given goal state. After training, the *achiever* can be deployed as a goal-reaching policy without further tuning. Authors also spent time investigating how to design the loss function for the *achiever*. In the experiments, three new benchmarks are introduced and the proposed method is shown to outperform baselines.


**Limitations And Societal Impact:**

Yes

**Main Review:**

**Originality**: The main idea is novel and incorporates many sound ideas in the field including Dreamer, temporal distance, and disagreement for exploration. The related works are adequately cited and described so the contribution is clear to me. The questions asked from line 75-79 also help me understand how this work is different from previous contributions.

**Quality**: Overall, the main idea kinda makes sense. However, some details still bother me:

- Line 176-177, “... to incorporate learning signals from far-away goals, we include them by sampling images from a different trajectory.” -> I am not sure whether images from different trajectories are necessary *far away*. Theoretically, they can be close to each other in terms of “the number of time steps between them”. Does this implementation cause any training unstability?

The technical contributions are supported by the ablation study shown in Figure 8. Thank the authors for running this experiment, I find it pretty informative.

**Clarity**: The high-level idea is presented clearly in the Introduction, Figure 1, and Figure 2. The low-level implementation details are described thoroughly in the supplementary material (which contains code). However, the mathematical notations in Section 2 are confusing and I hope authors can clean them up. I have the following questions:

- What is the capital $V$ in Figure 2? Is it the same as the value $v$ in the text?
- In section 2.1, $e_t$ shows up in equation 1's Posterior term but is not described in the text so its appearance hurts the readability.
- In section 2.2, I think the reward function (equation 3 & 4) of the explorer should appear before equation 2 for readers to have a better understanding of what $v^e$ should look like. Currently, $v^e$ shows up abruptly in equation 2 and its relationship with $r^e$ is only described near the end of section 2.2.
- Line 113 "produced by the encoder $q_{\phi}$", so both $q_{\phi}$ and $enc_{\phi}$ in equation 1 are called "encoder"? I think authors are probably referring to the posterior estimator as an "encoder" following the VAE literature but it would be nice if authors can clarify its relationship with another "encoder" $enc_{\phi}$ in the revised manuscript.
- In equation 3, what’s the relationship between $\hat{z}$ and $z$ defined at line 97?
- Line 146 "... to the goal $e_g$". It would be nice if authors describe it as "the embedding of the goal $e_g$” otherwise it confuses me as authors also denote g as goal at line 11 & 12 of Algorithm 1.
- In equation 7, it's not clear why \hat notation is introduced. How are variables with \hat different from their original meaning? Also, why is $\approx$ used instead of $=$?

Some other wordings that confuse me:

- Line 272-273, “real world data from the replay buffer”, it would be more clear if “real world data” is substituted with something like observed data as all the results in this paper come from the simulation.

Many obvious typos appear in the text:

- Line 31, "a traditional RL would have to" -> "a traditional RL agent would have to"
- Line 61, "between an the states" -> "between the states"
- Line 71-72, "and thus enormous computational ..." -> "and thus consume enormous computational ..."
Line 120, “seek out situations are …” -> “seek out situations that are”.
Line 156, “We address this is by …” -> “We address this by ...”.
Line 250, “train a train a “ -> “train a”.
Line 265, “i.e whether ..,” -> “i.e., whether”
Line 267, “that is need” -> “that is needed”

**Significance**: Since all the experiments are run on three *new* benchmarks introduced by authors, it’s really hard to tell whether the results are important. As the proposed idea is not limited to vision-based observations, I wonder whether authors can show results on state-based environments used by [1] and compare the performance. Experiments on state-based environments will help us understand 1) whether imagination is needed for low-dimensional observation space and 2) whether the proposed method, being more complicated, accidentally fails for simpler domains.

[1] Dynamical Distance Learning for Semi-Supervised and Unsupervised Skill Discovery, Hartikainen et al.

---

I am willing to raise my scores if authors addressed my concerns and above.

**Time Spent Reviewing:**

5

---

> ### Author Response · Authors · 2021-08-11
> **Author Response: provided significant new experiments on prior benchmarks**
>
>
> Thank you for thorough and helpful comments! We have updated the paper with the clarity suggestions, and also performed significant additional comparisons on prior benchmarks to address the experimental evaluation concerns. We would like to further stress that our proposed benchmark is significantly more complex and more informative for evaluation than those used in prior work, and we believe that the empirical performance of our agent on this benchmark constitutes a significant advance of the prior capabilities of unsupervised agents (see the details on the supplementary website: https://sites.google.com/view/exanet/home). We detail how we addressed the reviewer’s concerns below. If there are any further concerns that prevent you from accepting the paper, please let us know and we will attempt to address them!
>
> > “all the experiments are run on three new benchmarks introduced by authors”
>
> Our original submission contained results on a new and significantly more challenging benchmark than those considered by prior work, since most of the prior benchmarks have saturated performance and are less useful for designing new methods. However, we agree that a comparison on these prior benchmarks strengthens the paper. Therefore, we have performed comparisons on 15 additional environments taken from the SkewFit, Plan2Explore, and DISCERN papers and several baselines. Our agent performs comparably or better than all previous methods on these environments, without us changing any of its hyperparameters.
>
> ### Evaluation on tasks from SkewFit [ICML 2020]
>
> We report the distance to the goal following the SkewFit evaluation protocol. Our agent significantly outperforms prior work. We further note that this benchmark is largely solved and future work should focus on harder benchmarks such as the one proposed in our paper.
>
> |               | RIG   | RIG + Hazan et al. | RIG + HER | DISCERN | RIG + AutoGoal GAN | RIG + DISCERN-g | RIG + # Exploration | RIG + Rank-Based Priority | Skew-Fit | Ours  |
> |---------------|-------|--------------------|-----------|---------|--------------------|-----------------|---------------------|---------------------------|----------|-------|
> | Visual Pusher | 0.077 |              0.059 |     0.075 |   0.094 |              0.088 |            0.07 |               0.088 |                     0.067 |    0.049 | **0.023** |
> | Visual Pickup | 0.037 |              0.039 |     0.035 |   0.039 |              0.039 |           0.032 |                0.04 |                     0.035 |    0.018 | **0.014** |
>
> ### Evaluation on tasks from Plan2Explore [ICML 2020]
>
> Plan2Explore can adapt to new tasks but it needs the reward function to be known at test time while our approach does not require any access to rewards. To compare on the same benchmark, we create goal images that correspond to the reward functions. Note that this setup is arguably harder for our agent, but is much more practical. Our agent outperforms Plan2Explore on most tasks and even performs comparably to state of the art oracle agents (DrQ, DrQv2, Dreamer) that need task-specific reward signal even during training.
>
> |                         | DrQ | DrQv2 | Dreamer | Plan2Explore | Ours |
> |-------------------------|-----|-------|---------|-----|------|
> | zero-shot               | ❌   | ❌     | ❌       | ✅   | ✅    |
> | walker stand            | 957 |   968 |     977 | 331 | **957**  |
> | hopper stand            | 930 |   957 |     923 | **841** | **840**  |
> | cartpole balance        | 973 |   989 |     979 | **950** | 886  |
> | cartpole balance sparse | 983 |   983 |     941 | 860 | **996**  |
> | pendulum swingup        | 383 |   837 |     833 | **792** | **788**  |
> | cup catch               | 962 |   909 |     962 | **962** | **969**  |
> | reacher hard            | 392 |   970 |     817 |  66 | **937**  |
>
> ### Evaluation on tasks from DISCERN [ICLR 2019]
>
> The authors have not released their environments to the public. We were not able to reproduce the evaluation procedure from DISCERN despite contacting authors and following exact instructions in the original paper. Nevertheless, we show the goal completion percentage results obtained with our reproduced evaluation compared to DISCERN results from the original paper below. We see that our agent performs adequately on this benchmark and appears to be competitive. We further suspect that the goals that we generated are harder than the ones used in the original paper, such as in the cartpole environment where our goals require swinging the pole higher up.
>
> |             | DISCERN  | Ours     |
> |-------------|----------|----------|
> | ball in cup | 76.5% | **84%**       |
> | cartpole    | 21.3%     | **35.9%** |
> | finger      | 21.8%     | **40.9%** |
> | pendulum    | 75.7%     | **79.1%** |
> | pointmass   | 49.6% | **83.2%**   |
> | reacher     | 87.1% | **100%**     |
>
> ### Our challenging benchmark
>
> - These additional experiments above show that these previous tasks are largely solved (and in some cases are difficult to replicate) and are not suitable for developing new agents. This is the key motivation for complex benchmarks in our paper with 4 different environments and over 40 goals in total which feature higher diversity, longer horizons, and higher-dimensional agents than was considered previously.
> - Furthermore, note that we did not create new environments but added goal images for test time evaluation in environments that have been used by the community. For instance, the Kitchen environment that we show results on has been used by many prior papers but no prior work has accomplished successful results without the use of expert demonstrations. We believe our results as well as benchmark are valuable contributions to the reinforcement learning community.
>
> > “results on state-based environments”
>
> We agree that it would be instructive to evaluate our agent from states and compare to prior state-based benchmarks. We have started implementing our agent from states, but significant changes to the code are required since we primarily focus on image-based RL. We will include state-based evaluation in the final paper. However, note that we provide multiple additional results on prior benchmarks in the general response.
>
> > “I am not sure whether images from different trajectories are necessary far away…  Does this implementation cause any training unstability?”
>
> We did not observe any instability or performance degradation, as also shown in Fig 8. This is consistent with literature on contrastive learning, where the negative samples can include samples that are similar to the positive samples. Images that are close in terms of the number of time steps would appear much more often in the positive samples than in the negative samples, so the learned distance will be small on them. For images that are further away, the negative samples will ensure that the predicted distance will be a large constant, rather than an arbitrary value that the network was never trained on.
>
> > “the mathematical notations in Section 2 are confusing”
>
> Unfortunately, we are not allowed to update the manuscript to clarify these issues, so we attempt to clarify them here. We have updated our internal version of the paper to address the issues pointed out by the reviewer.
> - Capital V is the value of a particular state, i.e. $V_t = v_t(s_t, e_g)$
> - L113, this is a typo, $q_{phi}$ will be called a posterior.
> - $\hat{z}$ is a deterministic prediction of an ensemble network \theta^k. This prediction is trained to match z with the MSE objective.
> - $\hat{e}$ is an embedding predicted back from the latent state s, while e is the embedding directly encoded from x. We train $\hat{e}$ to match e, which is denoted with $\approx$

---

> > ### Comment · Reviewer_bDZB · 2021-08-25
> > **Re: Rebuttal**
> >
> > Thanks for all the experiments. I've updated my score to be weak accept.

---

> ### Author Response · Authors · 2021-08-17
> **Discussion Period**
>
> Dear Reviewer,
>
> Hope you got a chance to read our rebuttal. Please let us know whether you have any further concerns remaining that prevent you from accepting the paper.

---

### Official Review · Reviewer_txvK · 2021-07-19

**Rating:** 6
**Confidence:** 4

**Summary:**

This paper proposes a method to learn goal-conditioned policies by combining ideas from goal-conditioned RL and model-based exploration. It operates by training an exploration policy within the model to maximize model disagreement (as proposed in previous works), as well as a goal-conditioned policy to reach states in the replay buffer. These are then both deployed to grow the replay buffer, and the process is repeated. The approach is evaluated on a newly-defined set of benchmark tasks based on the Deepmind Control suite, as well as simulated robotic manipulation. They report better performance compared to previous methods such as Skew-Fit, DIAYN and two others. They also investigate two different choices of distance functions for the goal-conditioned RL component, and show that the optimal one depends on the setting.

**Limitations And Societal Impact:**

Yes.

**Main Review:**

Update after the rebuttal: I thank the authors for having added an extensive comparison to prior work on previous environments. Since this was in my view the main limitation of this work, I have raise my score from a 4 to a 6.


============================================================================

Methods-wise, this is essentially a combination of model-based exploration methods which use the disagreement between dynamics models to incentivize exploration, and goal-conditioned RL (which is performed inside the dynamics model here). Incidentally, a piece of related work which should be mentioned is the following https://arxiv.org/pdf/1911.00617.pdf which proposed model-based exploration with multiple dynamics models and also provided some theoretical results for this class of algorithm.

The novelty of this paper is somewhat limited, in that it is a fairly straightforward combination of existing techniques. However I am not aware of this specific combination being investigated before.

A big issue with the experimental setup is that the tasks are new and defined in this paper. It’s ok to define new tasks to see where existing methods fail, but it’s also important to know how the proposed method performs on tasks where existing methods are known to succeed. Also, checking that the baselines work in settings where they are known to succeed confirms the implementation, hyperparameter tuning etc is correct. Please include results for tasks from the papers where the baseline methods were proposed.

For example:
- the simple grid-world environment from the Skew-Fit paper and/or the 2d navigation one from DIAYN. Both of those methods should work there. How do they compare to the proposed method in terms of sample complexity?
- DIAYN uses the learned policy to initialize a policy optimizing task reward on some Mujoco tasks. Please include a comparison of the proposed method with the same setup.
- etc

It’s also a bit unsatisfying that you need different distance metrics for different types of task (i.e. cosine for RoboYoga and temporal for the robotic manipulation).

It is unclear from the plot in Figure 4 if the other methods (DIAYN, DDL, SkewFit, GCSL) fail to learn well or if they are slower to converge. Please include all their final performance after longer training (could be a dashed line).

It is unclear why step 8 in the algorithm is necessary, since in principle (according to MAX, Plan2Explore etc) the disagreement should be sufficient for exploring the different regions of the state space. The way \pi^g is trained is similar to Skew-Fit without the skewing step (and also training it in the model rather than the real environment), which that paper reports does not explore very fast. A useful ablation would be to try removing step 8 in Algorithm 1 and see how well it works. Does \pi^g discover different types of states than \pi^e? It seems like both step 7 and 8 are meant to add novel states to the buffer, but it’s not clear if they are exploring differently and are both needed.

Minor:
Lines 7-8: “Unlike prior methods that explore by reaching previously visited states, our explorer…” this is not true, there are several methods (MAX, Neural-E3, Plan2Explore) which explore by planning/training a policy inside the model.
Line 32: “A traditional RL would” -> a traditional RL agent would
Line 66: The introduce -> We introduce
Line 234: nvolves -> involves

**Time Spent Reviewing:**

3.5

---

> ### Author Response · Authors · 2021-08-11
> **Author Response: provided significant new experiments on prior benchmarks**
>
> Thank you for your constructive and insightful feedback! In our original submission, we evaluated our method on a new benchmark designed to be much harder than tasks considered in prior work (see the details on the supplementary website: https://sites.google.com/view/exanet/home). We agree that evaluation on prior benchmarks is helpful to put our results in context, and we have now additionally evaluated on 15 environments from prior work as outlined below, including environments from SkewFit as requested. Our agent performs comparably or better than all previous methods on these environments, without us changing any of its hyperparameters.  Please let us know whether this addresses all your concerns or whether there are any remaining points – that cause you to continue recommending your current score – that we can address!
>
> >  “important to know how the proposed method performs on tasks where existing methods are known to succeed”
>
> Our original submission contained results on a new and significantly more challenging benchmark than those considered by prior work, since most of the prior benchmarks have saturated performance and are less useful for designing new methods. However, we agree that a comparison on these prior benchmarks strengthens the paper. Therefore, we have performed comparisons on 15 additional environments taken from the SkewFit, Plan2Explore, and DISCERN papers and several baselines. Our agent performs comparably or better than all previous methods on these environments, without us changing any of its hyperparameters.
>
> ### Evaluation on tasks from SkewFit [ICML 2020]
>
> We report the distance to the goal following the SkewFit evaluation protocol. Our agent significantly outperforms prior work. We further note that this benchmark is largely solved and future work should focus on harder benchmarks such as the one proposed in our paper.
>
> |               | RIG   | RIG + Hazan et al. | RIG + HER | DISCERN | RIG + AutoGoal GAN | RIG + DISCERN-g | RIG + # Exploration | RIG + Rank-Based Priority | Skew-Fit | Ours  |
> |---------------|-------|--------------------|-----------|---------|--------------------|-----------------|---------------------|---------------------------|----------|-------|
> | Visual Pusher | 0.077 |              0.059 |     0.075 |   0.094 |              0.088 |            0.07 |               0.088 |                     0.067 |    0.049 | **0.023** |
> | Visual Pickup | 0.037 |              0.039 |     0.035 |   0.039 |              0.039 |           0.032 |                0.04 |                     0.035 |    0.018 | **0.014** |
>
> ### Evaluation on tasks from Plan2Explore [ICML 2020]
>
> Plan2Explore can adapt to new tasks but it needs the reward function to be known at test time while our approach does not require any access to rewards. To compare on the same benchmark, we create goal images that correspond to the reward functions. Note that this setup is arguably harder for our agent, but is much more practical. Our agent outperforms Plan2Explore on most tasks and even performs comparably to state of the art oracle agents (DrQ, DrQv2, Dreamer) that need task-specific reward signal even during training.
>
> |                         | DrQ | DrQv2 | Dreamer | Plan2Explore | Ours |
> |-------------------------|-----|-------|---------|-----|------|
> | zero-shot               | ❌   | ❌     | ❌       | ✅   | ✅    |
> | walker stand            | 957 |   968 |     977 | 331 | **957**  |
> | hopper stand            | 930 |   957 |     923 | **841** | **840**  |
> | cartpole balance        | 973 |   989 |     979 | **950** | 886  |
> | cartpole balance sparse | 983 |   983 |     941 | 860 | **996**  |
> | pendulum swingup        | 383 |   837 |     833 | **792** | **788**  |
> | cup catch               | 962 |   909 |     962 | **962** | **969**  |
> | reacher hard            | 392 |   970 |     817 |  66 | **937**  |
>
> ### Evaluation on tasks from DISCERN [ICLR 2019]
>
> The authors have not released their environments to the public. We were not able to reproduce the evaluation procedure from DISCERN despite contacting authors and following exact instructions in the original paper. Nevertheless, we show the goal completion percentage results obtained with our reproduced evaluation compared to DISCERN results from the original paper below. We see that our agent performs adequately on this benchmark and appears to be competitive. We further suspect that the goals that we generated are harder than the ones used in the original paper, such as in the cartpole environment where our goals require swinging the pole higher up.
>
> |             | DISCERN  | Ours     |
> |-------------|----------|----------|
> | ball in cup | 76.5% | **84%**       |
> | cartpole    | 21.3%     | **35.9%** |
> | finger      | 21.8%     | **40.9%** |
> | pendulum    | 75.7%     | **79.1%** |
> | pointmass   | 49.6% | **83.2%**   |
> | reacher     | 87.1% | **100%**     |
>
> ### Our challenging benchmark
>
> - These additional experiments above show that these previous tasks are largely solved (and in some cases are difficult to replicate) and are not suitable for developing new agents. This is the key motivation for complex benchmarks in our paper with 4 different environments and over 40 goals in total which feature higher diversity, longer horizons, and higher-dimensional agents than was considered previously.
> - Furthermore, note that we did not create new environments but added goal images for test time evaluation in environments that have been used by the community. For instance, the Kitchen environment that we show results on has been used by many prior papers but no prior work has accomplished successful results without the use of expert demonstrations. We believe our results as well as benchmark are valuable contributions to the reinforcement learning community.
>
> >  "How do prior methods compare to the proposed method in terms of sample complexity?(on environments from prior works)"
> - On the visual pickup skew-fit environments our approach takes about the same number of samples as skewfit to meet skewfit’s reported performance, whereas on the skewfit push environment our approach requires about 25% fewer samples. On the DISCERN tasks our method is 100x more sample efficient.
>
> > “DIAYN uses the learned policy to initialize a policy optimizing task reward .. include comparison of the proposed method with the same setup”
> - The problem setting we consider doesn’t have a task reward provided for a test task, but rather just a goal image. We use this harder setup since it’s closer to the eventual real-world setting in which it is infeasible to specify task reward.
>
> >  “A useful ablation would be to try removing step 8 in Algorithm 1 and see how well it works”
> - We ran the suggested ablation, where we use only the explorer policy for data collection. See the results at this link: https://drive.google.com/file/d/1LdPXUkrkjQtGFuONMoCshYJtA1O4slzF/view?usp=sharing. We find that the achiever data collection is actually not necessary with our final agent. In our preliminary experiments, the achiever data collection was important, but this experiment shows that the final agent is robust enough to perform well even with only explorer data collection. Nevertheless, we believe future work may use achiever data collection since it might be necessary on some tasks, and it does not require rewards so it can be performed entirely in an unsupervised way.
>
> > “a piece of related work which should be mentioned is the following https://arxiv.org/pdf/1911.00617.pdf ...”
> -  We thank the reviewer for pointing this out, and this work will be added to the related works section.
>
> > “The novelty of this paper is somewhat limited”
> -  We emphasize that our major contribution is to investigate the importance of exploration for goal-conditioned learning (GCRL). Different from prior work in GCRL, our agent does not simply revisit goals from the replay buffer, but attempts to discover novel goals through multi-step imagination. We also note that while some prior work considered forward-looking exploration, our paper is the first to do so while not using demos, rewards or states at neither train nor test time, thus being entirely unsupervised. Our empirical results further show that such forward-looking exploration is crucial for good performance in GCRL, our paper being both the first to investigate forward-looking exploration for GCRL and the first to empirically achieve good exploration on complex environments such as kitchen manipulation.
>
> > "there are several methods (MAX, Neural-E3, Plan2Explore) which explore by planning/training a policy inside the model"
> - We will clarify that our main novel contribution is to use forward-looking exploration for goal discovery. Prior work in goal discovery focused on very different exploration strategies like oversampling certain goals from the replay buffer and reaching them, while our approach is qualitatively different. Further, while some prior work considered forward-looking exploration, our paper is the first to do so while not using demos, rewards or states at neither train nor test time, thus being entirely unsupervised.

---

> > ### Author Response · Authors · 2021-08-31
> > **Request for follow up (deadline in 2 days)**
> >
> > Dear Reviewer,
> >
> > Just gently checking once again to see if you had an opportunity to look at our rebuttal. We hope to have addressed all your concerns but if you have any more questions that prevent you from accepting the paper, please let us know as the deadline is just two days away.

---

> > > ### Comment · Reviewer_txvK · 2021-09-03
> > > **Thank you for the additional experiments.**
> > >
> > > Thank you for the additional experiments, they address my main concern with the paper. I have raised my score accordingly.

---

> ### Author Response · Authors · 2021-08-16
> **Discussion period**
>
> Dear Reviewer,
>
> Hope you got a chance to read our rebuttal. Please let us know whether you have any further concerns remaining that prevent you from accepting the paper.

---

### Author Response · Authors · 2021-08-11
**Summary of updates: provided significant new experiments on prior benchmarks**

We would like to thank all reviewers for constructive feedback! To summarize, we propose an effective method that discovers goals with forward looking-exploration and learns to achieve them in a zero-shot manner from high-dimensional image input without using rewards or states at neither training nor test time.

All the reviewers suggested additional experiments, in particular, comparison to prior methods on the benchmarks used in those papers. We are pleased to report that we have performed comparisons on 15 additional environments taken from the SkewFit, Plan2Explore, and DISCERN papers and several baselines. Our agent performs comparably or better than all previous methods on these environments, without us changing any of its hyperparameters. Note that the benchmark we proposed in the main paper is substantially more challenging than these prior tasks/environments where we outperformed these methods by a significant margin.

Below, we provide a recap summary of only the key questions, and refer to individual reviewer replies for detailed rebuttal.

## Evaluation on tasks from SkewFit (ICML 2020)

We report the distance to the goal following the SkewFit evaluation protocol. Our agent significantly outperforms prior work. Pushing and picking up blocks from visual observations is largely solved, so future work will likely focus on harder benchmarks such as the one proposed in our paper.

|               | RIG   | RIG + Hazan et al. | RIG + HER | DISCERN | RIG + AutoGoal GAN | RIG + DISCERN-g | RIG + # Exploration | RIG + Rank-Based Priority | Skew-Fit | Ours  |
|---------------|-------|--------------------|-----------|---------|--------------------|-----------------|---------------------|---------------------------|----------|-------|
| Visual Pusher | 0.077 |              0.059 |     0.075 |   0.094 |              0.088 |            0.07 |               0.088 |                     0.067 |    0.049 | **0.023** |
| Visual Pickup | 0.037 |              0.039 |     0.035 |   0.039 |              0.039 |           0.032 |                0.04 |                     0.035 |    0.018 | **0.014** |

## Evaluation on tasks from Plan2Explore (ICML 2020)

Plan2Explore can adapt to new tasks but it needs the reward function to be known at test time while our approach does not require any access to rewards. To compare on the same benchmark, we create goal images that correspond to the reward functions. Note that this setup is arguably harder for our agent, but is much more practical. Our agent outperforms Plan2Explore on most tasks and even performs comparably to state of the art oracle agents (DrQ, DrQv2, Dreamer) that use true task rewards during training.

|                         | DrQ | DrQv2 | Dreamer | Plan2Explore | Ours |
|-------------------------|-----|-------|---------|-----|------|
| zero-shot               | ❌   | ❌     | ❌       | ✅   | ✅    |
| walker stand            | 957 |   968 |     977 | 331 | **957**  |
| hopper stand            | 930 |   957 |     923 | **841** | **840**  |
| cartpole balance        | 973 |   989 |     979 | **950** | 886  |
| cartpole balance sparse | 983 |   983 |     941 | 860 | **996**  |
| pendulum swingup        | 383 |   837 |     833 | **792** | **788**  |
| cup catch               | 962 |   909 |     962 | **962** | **969**  |
| reacher hard            | 392 |   970 |     817 |  66 | **937**  |

## Evaluation on tasks from DISCERN [ICLR 2019]

The authors have not released their environments to the public. We were not able to reproduce the evaluation procedure from DISCERN despite contacting authors and following exact instructions in the original paper. Nevertheless, we show the goal completion percentage results obtained with our reproduced evaluation compared to DISCERN results from the original paper below. We see that our agent performs competitively on this comparison. We further suspect that the goals that we generated are harder than the ones used in the original paper, such as in the cartpole environment where our goals require swinging the pole higher up.

|             | DISCERN  | Ours     |
|-------------|----------|----------|
| ball in cup | 76.5% | **84%**       |
| cartpole    | 21.3%     | **35.9%** |
| finger      | 21.8%     | **40.9%** |
| pendulum    | 75.7%     | **79.1%** |
| pointmass   | 49.6% | **83.2%**   |
| reacher     | 87.1% | **100%**     |

## Our challenging benchmark

- The additional experiments above show that these previous tasks are largely solved (and in some cases difficult to replicate) and are not suitable for developing new agents. This is the key motivation for introducing new challenging benchmarks in our paper with 4 different environments and over 40 goals in total which feature higher diversity, longer horizons, and higher-dimensional agents than was considered previously (see the details on the supplementary website: https://sites.google.com/view/exanet/home).
- Furthermore, note that we did not create new environments but added goal images for test time evaluation in environments that have been used by the community. For instance, the Kitchen environment that we show results on has been used by many prior papers *but no prior work has accomplished successful results without the use of expert demonstrations*. In contrast, we do not use demos, rewards, or states at neither train nor test time.

We believe our results as well as the benchmark are, in addition to our technical approach, also valuable contributions to the reinforcement learning community.

---

### Decision · Program_Chairs · 2021-09-27

**Decision:**

Accept (Poster)

**Comment:**

This paper proposes to learn a useful goal-conditioned policy in the absence of reward by jointly training an explorer that learns to visit uncertain states and an achiever that learns to reach a given goal state. All of the reviewers agreed that this is a novel combination of the prior work on learning a goal-conditioned policy in self-supervised learning and model-based RL, though each component is not entirely new. The only initial concern was that the proposed method was not evaluated on domains where the baselines were evaluated. However, the authors provided an additional result during the rebuttal period, and the majority of the reviewers are satisfied with it. Therefore, I recommend accepting this paper and suggest the authors to include the new result in the camera-ready version.